# Streamlined analysis of drug targets by proteome integral solubility alteration indicates organ-specific engagement

Tanveer Singh Batth [1,3] ✉, Marie Locard-Paulet[1,2,3], Nadezhda T. Doncheva [1], Blanca Lopez Mendez [1], Lars Juhl Jensen [1] & Jesper Velgaard Olsen [1] ✉

Proteins are the primary targets of almost all small molecule drugs. However, even the most selectively designed drugs can potentially target several unknown proteins. Identification of potential drug targets can facilitate design of new drugs and repurposing of existing ones. Current state-of-the-art proteomics methodologies enable screening of thousands of proteins against a limited number of drug molecules. Here we report the development of a label-free quantitative proteomics approach that enables proteome-wide screening of small organic molecules in a scalable, reproducible, and rapid manner by streamlining the proteome integral solubility alteration (PISA) assay. We used rat organs ex-vivo to determine organ specific targets of medical drugs and enzyme inhibitors to identify drug targets for common drugs such as Ibuprofen. Finally, global drug profiling revealed overarching trends of how small molecules affect the proteome through either direct or indirect protein interactions.

The primary objective in small molecule drug design is to maximize the potency and specificity towards the target protein. This ensures that the desired pharmacological effects through hypothesized mechanisms of action (MoAs) are achieved while minimizing the risk of off-target interactions that can lead to unforeseen and undesirable side effects. It has been suggested that most drugs exert their phenotypic effects through a variety of unknown MoAs[1]. Numerous techniques and strategies have been devised to comprehensively identify all potential protein targets, both intended ('ON') and unintended ('OFF'), of small molecule drugs.

Indirect approaches, such as those based on genetic screens using siRNA knockdowns[2], CRISPR knockouts[3] or yeast two-hybrid systems[4], offer versatility and straightforward readouts. However, they tend to produce higher rates of false positives due to their reliance on indirect detection methods, making the elucidation of MoAs more challenging. Direct methods, including those based on affinity purification with immobilized drugs and competitive activity chromatography, aim to unveil direct interactions between immobilized small molecules and

their target proteins or protein complexes[5]. Other approaches capitalize on changes in protein structure, and thermal or solvent stability upon small molecule binding to identify direct and indirect protein targets[6–11]. Typically, these strategies are combined with quantitative mass spectrometry-based proteomics readouts to achieve comprehensive proteome coverage and target identification. Nevertheless, each approach possesses its unique strengths and weaknesses, and there is no single method or strategy capable of identifying all potential protein targets for a specific compound. Trade-offs must be considered with respect to method sensitivity, proteome depth, and throughput.

One method that has garnered substantial interest and adoption for cellular drug target deconvolution is thermal proteome profiling (TPP), which integrates the cellular thermal shift assay with mass spectrometry-based proteome profiling for target identification[8,9]. The core concept behind TPP is based on detecting changes in protein thermal stability when proteins engage with a ligand or small molecule[12]. This change is assessed through the determination of the

[1]Novo Nordisk Foundation Center for Protein Research, University of Copenhagen, Copenhagen, Denmark. [2]Institut de Pharmacologie et de Biologie Structurale (IPBS), Université de Toulouse, CNRS, Université Toulouse III - Paul Sabatier (UT3), Toulouse, France. [3]These authors contributed equally: Tanveer Singh Batth, Marie Locard-Paulet. ✉e-mail: t.batth@cpr.ku.dk; jesper.olsen@cpr.ku.dk

melting point, which is the temperature at which 50% of the initial protein quantity is denatured and becomes insoluble. The melting point is extracted from the melting curve, generated by gradually increasing the temperature in steps until the protein is completely denatured and becomes insoluble. Traditionally, thermal shift assays have been employed in drug discovery, using techniques such as differential scanning fluorimetry (DSF), although multiple variations of thermal shift assays are available. In the case of TPP, the melting curves for all identified proteins are determined by assessing the soluble proteome fraction following exposure to various temperatures. This is conventionally achieved through a multiplexed proteome analysis employing tandem mass tags (TMT). In this approach, the tryptic peptides from each soluble fraction that were subjected to different temperatures are labeled with different isobaric TMT reagents and combined. This process generates melting curves for peptides and proteins within a given TMT experiment[13,14]. Typically, TPP experiments involve approximately 8-15 distinct temperature points, spanning the range of 37 °C to 70 °C. As a result, melting curves for thousands of proteins can be simultaneously obtained within a single experiment.

In this study, we present a streamlined label-free workflow based on proteome integral solubility alteration (PISA) to address some of these limitations by automating and simplifying the experimental, sample preparation, data acquisition, and bioinformatics analysis procedures. Our workflow demonstrates an increased capability to correctly identify drug targets, both directly bound to the drug and indirectly affected by its presence. We applied this workflow to rat organs and screened 23 different compounds for target identification. Additionally, we have developed a bioinformatics pipeline tailored for the analysis of large datasets generated in this study and have identified and successfully validated a drug target for a commonly used medication as proof of concept.

## Results

### Temperature optimization

When comparing the melting point of a specific protein in the presence or absence of a drug under drug-treated conditions, significant differences in stability may arise from drug binding or consequent conformational changes induced through perturbation in protein-protein interactions. These observations suggest that the protein may serve as a potential target for the drug or be implicated in its downstream effects (Fig. 1A). In this article, we consider a protein as "targeted" or "engaged" by a drug if the protein is temperature stabilized or destabilized following drug treatment. TPP offers numerous advantages, including the capacity to identify both direct and indirect drug targets, along with its compatibility in cellular, in vivo, and cellular extract settings. While this technique provides the advantage of proteome-wide screening for potential drug targets, it does come with certain limitations. These include low throughput, high experimental complexity, and the challenges associated with downstream data analysis, which can constrain large-scale screening across various drugs and concentrations. Several adaptations have been developed to optimize extraction conditions[15,16] or integrate the underlying melting curve such as proteome integral solubility alteration (PISA)[17] (Fig. 1B).

To address some of the limitations of TPP, different variations have been developed. Some attempt to bypass the requirement for acquiring data for all temperature points for a melting curve altogether by integrating the peak. This involves pooling the soluble fraction from different temperature points and comparing global protein abundance between conditions rather than assessing the melting point[17] (Fig. 1B). Others focus on refining downstream data analysis strategies for melting point determination[18]. We analyzed three different TPP studies and found that the median melting point of all proteins across experiments conducted in cells and cellular extracts typically falls between 50-55 °C, with nearly 50% of all protein melting temperatures

falling within this range (Supplementary Fig. 1, Supplementary Data1). Based on this, we hypothesized that a PISA-type melting curve integration could be achieved using fewer temperatures within this range to yield a higher number of reliable results (see Fig. 1B). From the same sample set, we pooled the soluble fraction at 53 °C, 56 °C, and 59 °C (see Fig. 1C). We used elevated temperatures over the median to increase the likelihood of detecting temperature stabilized drug targets with higher statistical confidence. Although this could bias against targets that are temperature destabilized, we hypothesized that measuring changes in solubility at higher temperatures where the relative abundance of a target protein could drastically increase would increase the likelihood of identifying protein targets.

To test this hypothesis, we conducted a TPP experiment using the broad kinase inhibitor Staurosporine within a temperature range of 37 °C to 67 °C (see Fig. 1C). Staurosporine is often utilized to benchmark chemical proteomic methods and a key component for the "Kinobead" technology, which is based on competitive affinity enrichment of kinases[8,19–23]. We identified the proteins with significantly regulated melting points in the presence of 10 μM Staurosporine by comparing: (1) the melting points for the TPP experiments between control DMSO-treated and Staurosporine-treated replicates; (2) the differences in protein abundance between the two pooled samples for TPP-PISA (Fig. 2A). Using both TPP and TPP-PISA (Fig. 2B, C), we successfully identified kinase targets of Staurosporine (Supplementary Data 2). However, the integrated approach yielded >2x the number of kinase and non-kinase targets with better statistical significance (Fig. 2D) and higher fold changes compared to standard TPP (Fig. 2B–E, Supplementary Data 2). It should be noted that to make the analysis comparable, TPP was performed without peptide prefractionation, which is a crucial step in TMT-based TPP experiments to guarantee higher peptide detection numbers and precision in peptide quantification which can influence the sensitivity and specificity in Staurosporine target identification. Nonetheless, these results confirm that the PISA strategy, using a limited number of temperatures, results in a larger number of identified Staurosporine targets while substantially increasing the throughput of MS analysis, as all experiments can be conducted within a single TMT multiplexed run. Furthermore, this approach simplifies data analysis, as determining the complete melting curve is not necessary to identify Staurosporine targets.

### Label FREE quantitative analysis using data-independent acquisition

We sought to further improve the protocol by bypassing TMT peptide labeling for TPP-style experiments. Although TMT provides the benefit of multiplexing and low missing values, the technique requires additional labeling, pooling, and sample preparation cleanup steps which leads to a significant reduction in throughput. TMT data also suffer from lower peptide identification rates and the well-known phenomenon of batch effects that poses extra downstream data analysis challenges[24]. More specifically, it has the disadvantage that once an experimental dataset is acquired, it is not trivial to compare the data across different batches of samples and conditions. Although it is indeed possible to analyze up to 18 multiplexed samples within a single run using the latest version of TMT[25], in the context of large-scale compound screening, the number of samples could be severely limiting. This is further exacerbated when accounting for large compound libraries with controls and replicates, even for integrated TPP-type experiments where melting curve generation is not required. Subsequently, we decided to use a label-free quantitation (LFQ)-based data-independent acquisition (DIA) approach for mass spectrometry proteomics analysis of integrated TPP-PISA samples. Although DIA analysis requires independent measurement of each sample, it has been demonstrated to be compatible for melting curve fitting using the standard TPP workflow[26] and the latest developments in mass spectrometry hardware[27] and search algorithms[28] enable higher

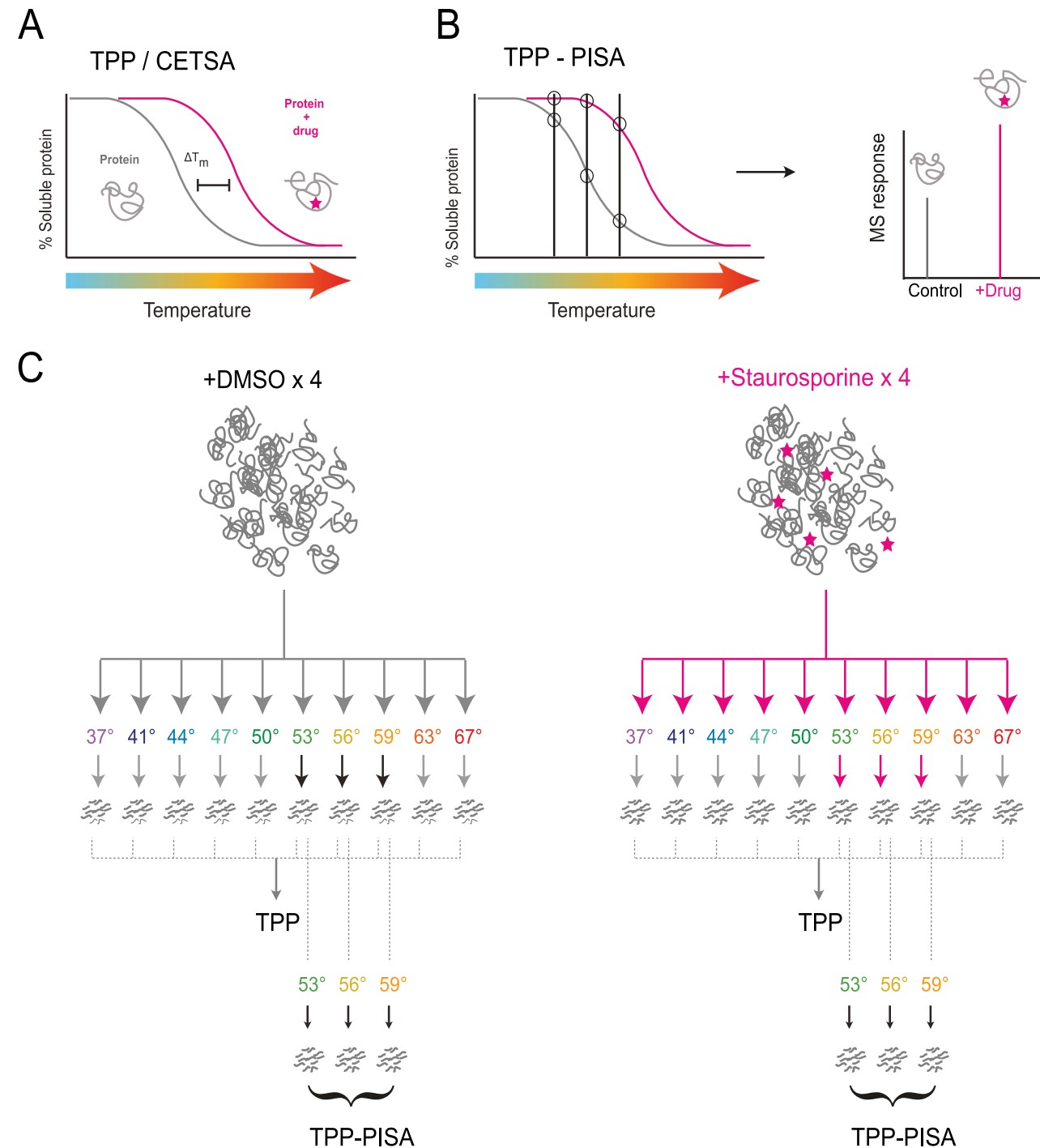

**Fig. 1 | TPP and optimized TPP-PISA analysis. A** Graphical illustration of the concept of thermal proteome profiling (TPP) or cellular thermal shift assay (CETSA) where a protein's drug interaction is determined through a melting curve that is generated based on the analysis of the soluble fraction after drug treatment followed by thermal heating at different temperatures. Differences in the melting point ($T_m$: the temperature at which only 50% of the protein is soluble) are used to determine whether a protein is a drug target. **B** Concept of the TPP-based method proteome integral stability assay (PISA). The melting curve is integrated by pooling only a few heat-treated fractions to analyze the relative quantity of soluble proteins. Consequently, a melting curve is not required to determine whether a protein is a drug target. **C** Illustration of the TPP and TPP-PISA experimental setup utilized to determine the effectiveness of profiling the soluble fraction over a temperature range (TPP, 37–67 °C) vs pooling a fewer number of temperature points (TPP-PISA, 53, 56, 59 °C). DMSO is used as control and compared against Staurosporine-treated protein extracts. Experiments are performed with 4 replicates each.

throughput of proteomic samples with high quantitative precision and fewer missing values. From our analysis, DIA-based TPP-PISA resulted in a larger number of identified proteins compared to TMT TPP and TMT TPP-PISA while utilizing only ~25% of the MS analysis time (Supplementary Fig. 2A). However, it must be noted that comparing protein groups across different search software can be difficult due to

differences in the assembly of protein groups. Nonetheless, DIA analysis also led to lower number of missing values across runs, with over 90% of the protein identifications containing intensities in all of the samples based on analysis of DMSO and Staurosporine-treated samples (Supplementary Fig. 2B). This was substantially higher compared to TMT TPP samples (~66%) and TMT TPP-PISA samples (~72%),

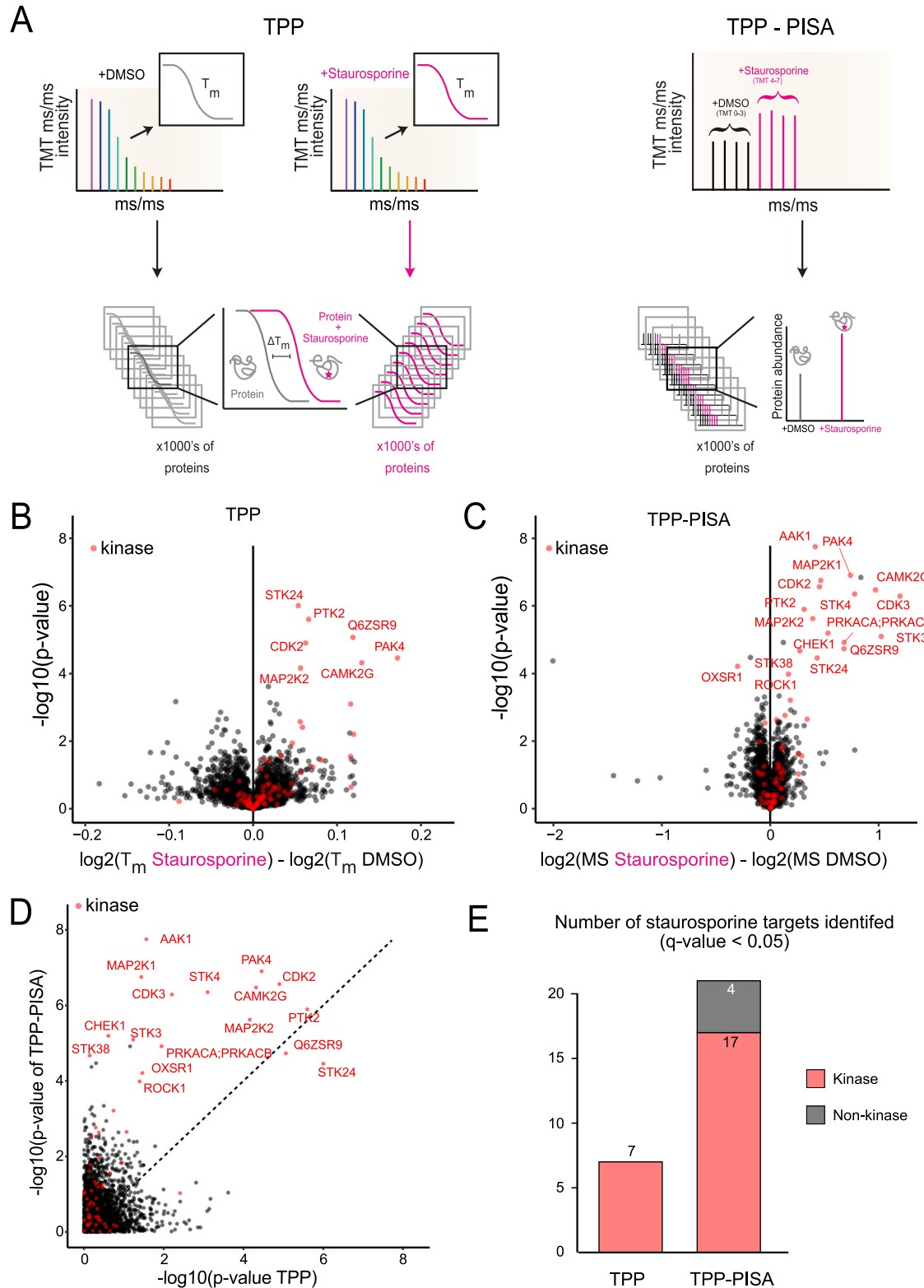

demonstrating the advantages of DIA over TPP for identifications, throughput, and data completeness (Supplementary Fig. 2C, D).

## Insoluble protein removal

One of the significant bottlenecks in TPP experiments involves laborious separation of the insoluble fraction from the soluble supernatant after heating at various temperatures. Traditionally, this is achieved through manual pipetting and transfer of the soluble portion to another tube or well, a time-consuming process that demands meticulous handling to avoid disturbing the insoluble fraction. Furthermore, the number of samples processed simultaneously is limited by the available positions in the centrifuge. Typically, centrifugation is conducted at speeds exceeding 20,000 × g for 30-60 minutes, and this limitation is exacerbated when using an ultracentrifuge as the number of samples that can be centrifuged simultaneously is typically less than 10. As a result, the scalability and throughput of TPP and similar experiments is substantially restricted.

**Fig. 2 | Comparison between TPP and optimized TPP-PISA. A** Data processing workflow for TPP and TPP-PISA are graphically presented on the left and right side, respectively. TPP requires the determination of melting curves which are compared across drug treatments to determine whether a protein is a potential drug target. TPP-PISA integrates the melting curve and thus only the difference in total protein abundance is required between drug treatment and control to determine whether a protein is a potential target. **B, C** Volcano plots showing the statistical output of the TPP (**B**) and TPP-PISA (**C**) experiments. −log10(*p*-values) are plotted on the vertical axis, and the log2-transformed differences between the Staurosporine- and DMSO-treated conditions are presented on the horizontal axis: differences of melting points ($T_M$) for (**B**), and differences between the corrected log2-transformed MS intensities of the TMT reporters for (**C**). Proteins on the right side of the volcano plot were stabilized by Staurosporine while proteins on the left side were destabilized. **D** Comparison of the *p*-values obtained for the protein groups identified in both TPP and TPP-PISA experiments. **E** Number of kinase and non-kinase protein groups identified in the two experiments with *q*-value < 0.05. See Supplementary Data 2 for additional technical replicates of the TPP-PISA experiment. The *q*-values were derived from *p*-values of two-sided unpaired t-tests (equal variance) that were corrected for multiple testing using false discovery rate[65].

To address this challenge, we explored the use of filter plates as an alternative to centrifugation for removing insoluble proteins after a TPP-PISA experiment as reported earlier[29]. Filter plates have been utilized for the removal of insoluble protein aggregates but their advantage over traditional centrifugation-based insoluble protein removal for TPP has not been thoroughly reported. Filter plates offer the capability to process up to 96 samples simultaneously, significantly reducing the time required for this step. Consequently, this also reduces pipetting time and errors as multi-channel pipettes or automated liquid transferring robots can be utilized for this crucial step. For instance, after just a 2-minute centrifugation, soluble proteins can be effectively separated and collected in a collection plate. To determine the extent of this advantage, we compared 30-minute centrifugation (with 20,000 × g relative centrifugal force) with 0.45 μm polytetrafluoroethylene (PTFE) filter plate on TPP-PISA samples treated with DMSO or Staurosporine (Fig. 3A, B, Supplementary Data 2). The 0.45 μm pore size is theoretically large enough for soluble proteins and protein complexes to pass through. The hydrophilic properties of the PTFE filter should prevent non-specific interactions, allowing soluble proteins to pass through while retaining large protein aggregates induced from heating of soluble protein mixtures. We also observed enhanced reproducibility when PTFE plates were used to remove insoluble proteins as demonstrated by the negative log10-transformation of the *p*-values for the kinase hits (Fig. 3C). Consequently, this strategy resulted in >2x the number of kinases and 7x higher number of non-kinase targets identified after Staurosporine inhibitor treatment when PTFE was utilized to remove insoluble proteins compared to centrifugation for 30 minutes (Fig. 3D).

## 96-well format workflow

Our optimizations of TPP have allowed us to establish a comprehensive 96-well plate-based experimental workflow (Fig. 4A). In this workflow, cellular extracts are distributed into 96-well PCR plates, and small molecule compounds are introduced into these extracts followed by incubation at 37 °C for 10 minutes. Subsequently, the cellular extracts are divided and transferred to different wells, each representing distinct temperature blocks for post-incubation heating. In our case, we employed three temperature blocks (at 53 °C, 56 °C, and 59 °C) for thermal profiling. Each block received one-third of the volume of the original compound-incubated extracts and was heated for four minutes (Supplementary Fig. 3). After heating, the contents of the temperature blocks are merged back into their original well and the well's content is transferred to a filter plate to remove insoluble proteins. The soluble proteins are collected into a new 96-well plate and are then prepared for overnight Lys-C/Trypsin protease digestion via protein aggregation on magnetic beads[30]. The samples are acidified after protease digestion and prepared for mass spectrometry analysis. Throughput was increased further via the transfer and loading of the overnight proteolytic digests directly into EvoTips in a 96-well plate format. EvoTips are disposable trap columns for the EvoSep One liquid chromatography (LC) system, which allow for the peptides storage and analysis[31]. This strategy bypasses the requirement for peptide elution from StageTips[32] or similar hydrophobic solid phase extraction that require additional steps of peptide loading, washing, and elution followed by evaporation of organic solvent and reconstitution in aqueous buffer prior to loading on the analytical column with a traditional liquid chromatography system. EvoSep One was coupled with the mass spectrometer operating in DIA mode and subsequent analysis of the raw data was facilitated by the DIA-NN software[28]. Consequently, the entire TPP and integrated TPP-PISA workflow can be automated in a 96-well format, significantly improving throughput and reproducibility.

## Evaluation of rat organ extracts

Finally, we assessed the suitability of rat organs as a source for TPP-PISA experiments. Rats serve as common animal models in preclinical pharmacological studies to investigate the toxicological effects of various drugs before human administration. Furthermore, TPP has been implemented in vivo with mouse models and whole blood[33], demonstrating the potential applicability of the method in vivo or ex vivo. While the phenotypic response to a drug in rats may not always mirror that observed in humans, rat studies still offer valuable pharmacological insights. Despite the inherent differences between humans and rats as species, they share 95% sequence similarity in orthologous protein-coding genes, suggesting that a small molecule targeting a human protein is likely to target the corresponding protein in rats[34]. We specifically used an Ensembl FASTA file of the rat proteome and mapped orthologous human proteins for each rat protein, which enabled us to determine targets for the human proteins. Using rat organs can broaden our proteome coverage covering different cell types and contexts and provide substantial quantities of cellular protein extracts that are often challenging to obtain from cultured human cell lines.

To evaluate this, we extracted the following organs from rats: liver, spleen, kidneys, quadriceps muscle, and two distinct brain regions: the hippocampus and cerebellum, in addition to whole brain samples (Fig. 4B). We anesthetized Wistar rats and perfused them with PBS saline to minimize blood contamination before organ removal and subsequent snap-freezing for native protein extraction. We assessed if our strategy was sensitive enough to identify protein melting differences in these organ extracts by treating extracts with Staurosporine for 10 minutes and applied our TPP-PISA pipeline. We observed a broad response in all organ extracts, except for those obtained from the whole brain (Supplementary Fig. 4A). This discrepancy could be attributed to the high lipid content in both white and gray matter of the brain, which could potentially interfere with temperature-based stability assays. Nevertheless, we obtained a response (as determined from the number of targets with a *q*-value ≤ 0.05) from the hippocampus regions of the rat brain. We also observed that a larger number of protein hits passing the statistical threshold for Staurosporine-treated extracts were not kinases, contrary to cell line analysis (Supplementary Fig. 4B, Supplementary Data 2). Furthermore, we found 133 different Staurosporine kinase targets across different rat organ extracts. The kinase targets covered 8 different kinase families (Fig. 4C), including 3 kinase targets which are considered atypical kinases. Different Staurosporine kinase target profiles were observed

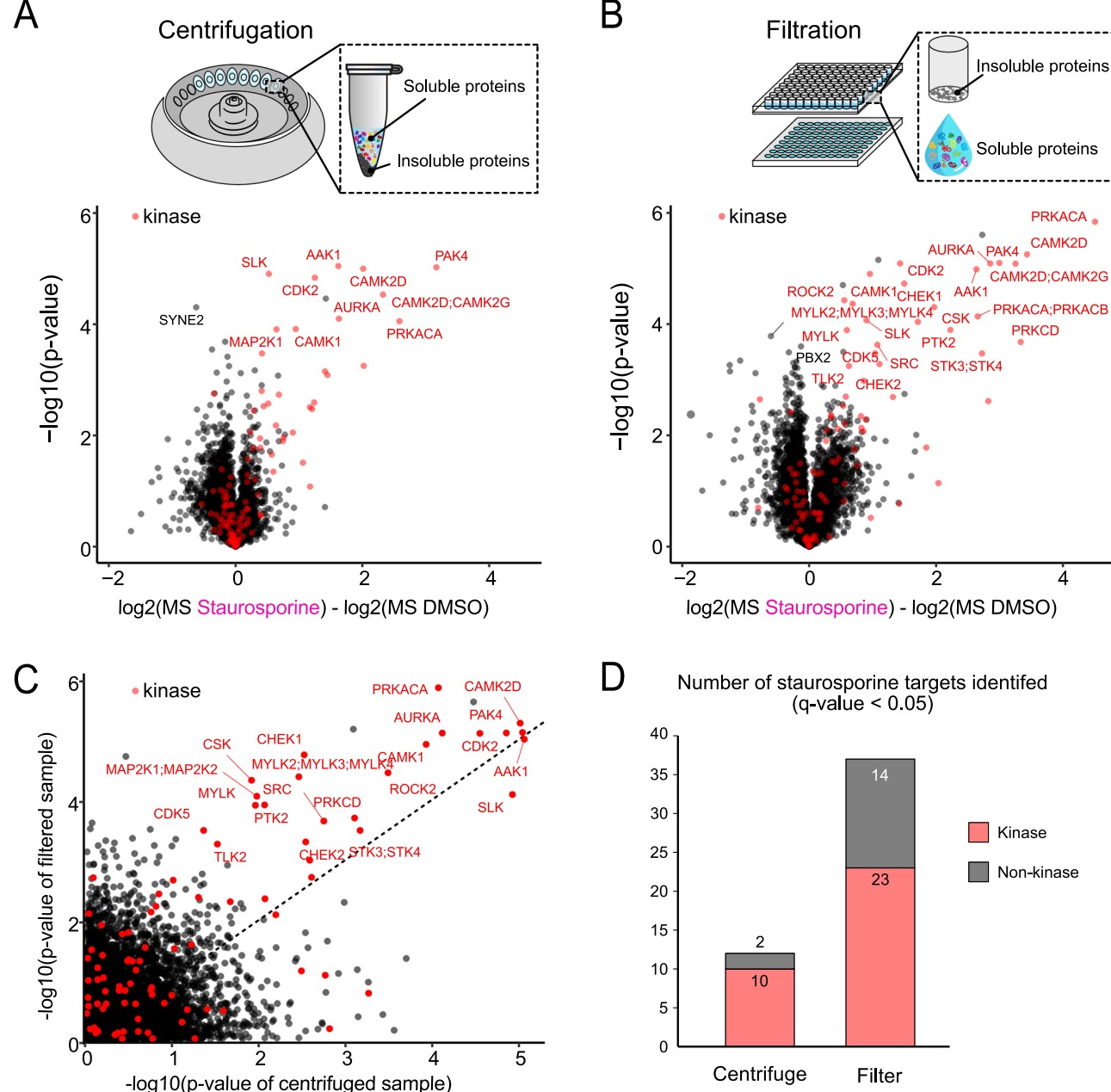

**Fig. 3 | Comparison between centrifugation and filtration for removal of insoluble proteins following temperature treatment. A, B** Volcano plots showing the statistical output of a TPP-PISA experiment using centrifugation (20,000 x *g* centrifugation for 20 minutes) (**A**) or filtration with a 0.45 μm pore size hydrophilic polytetrafluoroethylene filter plate followed by 2-minute centrifugation at 500 x *g* (**B**) for separating soluble and insoluble proteins. −log10(*p*-values) are plotted on the vertical axis, and the differences between the log2-transformed corrected MS intensities of the TMT reporters in the Staurosporine- and DMSO-treated conditions are presented on the horizontal axis. **C** Comparison of the *p*-values obtained for the protein groups identified in the two experiments presented in (**A**) and (**B**). **D** Number of Staurosporine kinase and non-kinase targets identified at *q*-value < 0.05 for the two experiments. The *q*-values were derived from *p*-values of two-sided unpaired t-tests (equal variance) that were corrected for multiple testing using false discovery rate[65].

across different rat organs (Supplementary Fig. 5). For example, we found kinases which were targeted by Staurosporine across rat organs, as well as kinase targets more exclusive to 1 or 2 organs (Fig. 4D, Supplementary Fig. 5). We also observed that some kinases could be temperature stabilized in one environment and temperature destabilized in another upon Staurosporine treatment (Fig. 4D, Supplementary Fig. 5).

**Drug screening**

We tested 22 compounds (excluding Staurosporine) spanning multiple clinical indications such as cancer, metabolic and neurological

diseases, and applied the complete workflow to different rat organ extracts as well as two human cell lines, HeLa and HepG2 (Supplementary Data 3). We developed a dedicated bioinformatics strategy to facilitate the identification of drug targets from large scale TPP-PISA type experiments. For a given sample, we compared protein quantities in a drug-treated condition with all the other conditions, relying on the hypothesis that two drugs will not impact the same protein melting points. To ensure the independence of control conditions, drugs were excluded from the control set of other drugs if they shared at least one common target according to the DrugBank[35] repository and/or if their pairwise structure similarities based on the Tanimoto index was ≥ 0.6

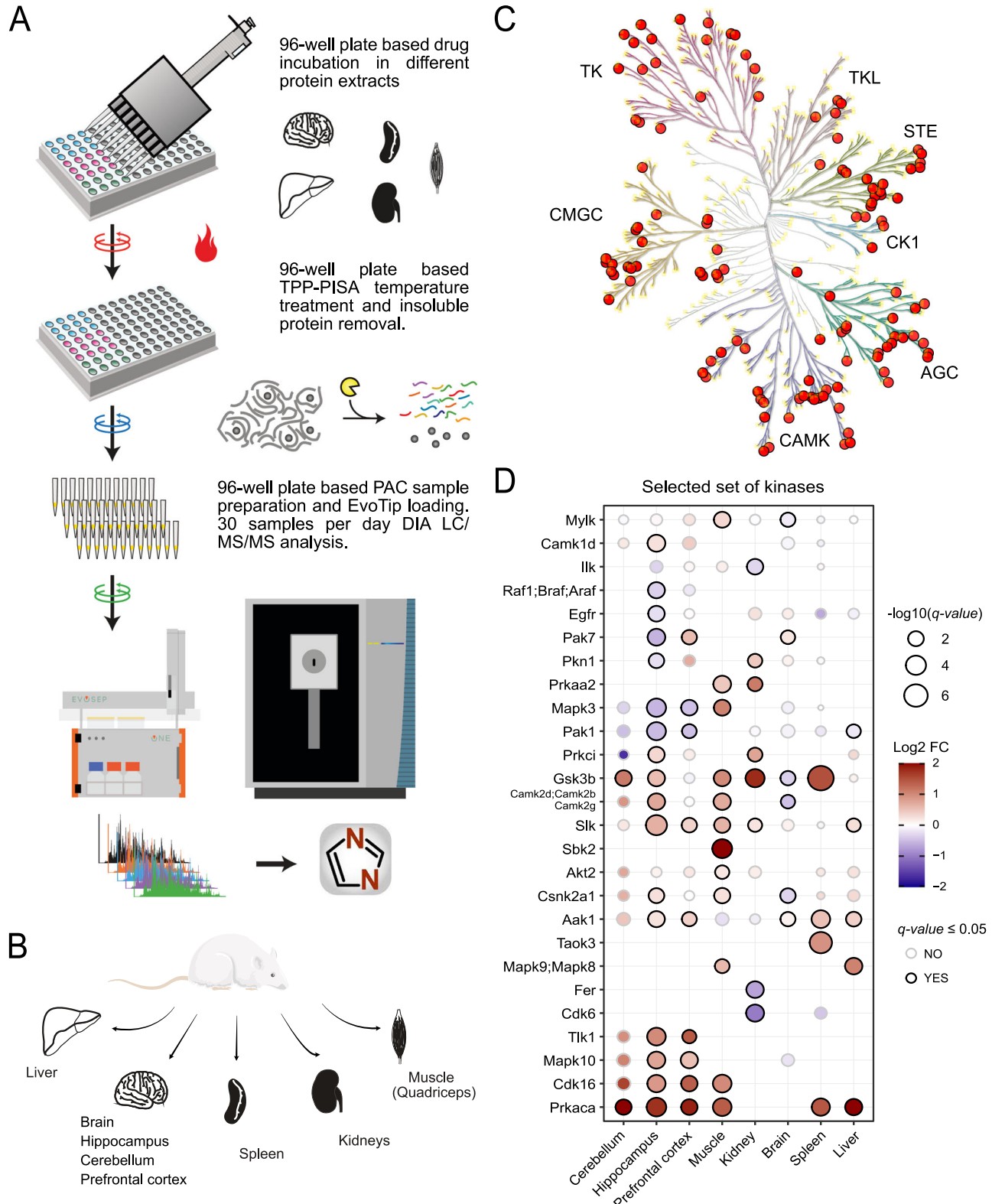

(Supplementary Data 4). We visualized this information in a drug similarity plot that shows the similarity of the compounds used in this study (Fig. 5A).

We applied this approach to analyze the entire dataset generated from screening different compounds across different rat organs and human cell lines. From this combined dataset, we identified an average of 4493 protein groups per experiment with a maximum of 666 protein targets across various rat organs and human cell lines for the 22

drugs profiled (Fig. 5B). We verified the specific targets of well-known compounds across different organs and cell lines, such as for Cobimetinib and SHP099, where Map2k1 (MEK1 protein kinase) and PTPN11 (Shp2 tyrosine phosphatase) consistently exhibited significantly increased temperature solubility in the presence of the respective drugs (Fig. 5D). Interestingly, we also observed that the extent of a drug's effect could vary across different proteome backgrounds. In other words, a protein target could be confirmed or observed for a

**Fig. 4 | Streamlined label-free TPP-PISA workflow and evaluation of rat organ extracts for TPP-PISA. A** 96-well plate-based workflow based on label-free MS analysis using data-independent acquisition method is illustrated. Briefly, protein extracts from different biological samples are aliquoted into a 96-well plate and compounds are added to the plate and incubated in a PCR machine at 37 °C. This is followed by splitting each well into three wells for heating at three different temperatures in the same PCR machine. After temperature treatment, the split aliquots are pooled into their original wells and transferred to a 96-well filter plate. Soluble proteins are collected in a collection plate where the samples are prepared using PAC protocol and overnight LysC/Trypsin digestion. Resulting peptides are loaded onto EvoTips and analyzed using DIA mass spectrometry. Data is searched using library-free DIA-NN 1.81 prior to downstream data analysis. **B** Organs extracted from Wistar rats are illustrated. **C** Staurosporine kinases targets found in rat organ extracts are mapped on the kinase phylogenetic tree as orange circles using KinMap[66]. **D** Representative dot plot of selected kinases identified as Staurosporine targets in rat organ extracts. Normalized fold change values log2(MS Staurosporine) −log2(MS DMSO) are color-coded for each point, and the size of the point is inversely proportional of the $q$-value. The stroke color for each point indicates whether the kinases pass the statistical threshold based on $q$-value. Full heatmap can be found in Supplementary Fig. 5.

drug in one set of cellular extracts, while the same protein might not be a target in a different extract, even if the protein was present. This observation implies that drugs may induce distinct responses in different samples, emphasizing the significance of considering diverse proteome contexts in TPP experiments. A notable observation was the prevalence of temperature destabilization in proteins following compound treatment, as opposed to stabilization. This trend was consistently observed across cell lines and rat organ extracts for most compounds (Fig. 5E).

We constructed protein–drug networks to visualize protein targets identified across different rat organ extracts and human cell lines. This enabled us to home in on high confidence targets for all the drugs tested in this study. For example, we mapped all identified protein targets of Cobimetinib found in different rat organs and human cell lines (Fig. 6A). Additionally, information about the nature of the protein–drug interaction was also represented, i.e. whether the identified target was found to be temperature stabilized or destabilized upon drug treatment in different biological backgrounds (see legend in Fig. 6A). Furthermore, shared protein drug targets between different drugs are also visualized as demonstrated in the Naproxen network where we added Ibuprofen and highlighted its targets shared with Naproxen (Fig. 6B). Lastly, we incorporated information about known protein targets (based on DrugCentral) as well as protein homology between any of the targets (previously known or hits in this study). This allowed us to identify targets of interest due to their homology to known targets even if the known target was not identified as a significant hit in our dataset. For example, the cytochrome P450 enzyme 3A4 (CYP2C9), which is a known Ibuprofen target (based on DrugCentral), is shown along with other homologous cytochrome P450 enzymes in the drug–target network of Ibuprofen (Fig. 6C). However, the known target CYP2C9 was not identified as a target in our analysis, nonetheless its homologs were found to be targets of Ibuprofen and thus part of this network. This visualization strategy highlights the advantage of constructing protein-drug networks from such an analysis.

Using this strategy, we found the protein Pirin (PIR) to be stabilized by Ibuprofen in many different proteome backgrounds suggesting a target of this drug (Fig. 7A). Like many common medications, the Ibuprofen MoA is not fully understood. Although it is hypothesized to inhibit cyclooxygenase enzymes 1 and 2, evidence for direct inhibition or other mechanism of actions have been proposed that suggest a complete understanding of its clinical effect remains elusive[36]. We utilized surface plasmon resonance (SPR) with recombinant human Pirin to experimentally evaluate the potential Ibuprofen-Pirin interaction suggested by the TPP-PISA dataset. SPR is a label-free technique commonly employed for the study of protein-ligand interactions and the screening of potential drug candidates. It excels in elucidating the specificity and selectivity of ligand binding while providing valuable insights into binding kinetics[37]. We used Nicotinamide and acetylsalicylic acid (commonly referred to as Aspirin) as negative controls. These were selected due to their similar small size, and both contain aromatic rings similar to Ibuprofen (Fig. 7B). To confirm binding to a recombinant version of human Pirin by SPR, we used a previously reported Pirin specific inhibitor Triphenyl Compound A (TPhA)[38] that binds Pirin with a $K_D$ of 0.7 μM as a positive control (Supplementary Fig. 6). Using SPR, we observed a dose-dependent response of Ibuprofen binding to Pirin (Fig. 7C) with a KD of 546 μM, although it can be difficult to accurately determine KD as the curve did not reach saturation. In contrast, the negative controls, Aspirin and Nicotinamide did not display a similar response (Fig. 7D). Furthermore, we tested the potential competitive binding between Ibuprofen and TPhA by repeating the SPR experiment for Ibuprofen in a running buffer containing 3 μM TPhA and observed that the Ibuprofen dose response was reduced. This would suggest that both molecules target the same site within Pirin (Fig. 7D). Contrarily, we found the beta-hexosaminidase (HexB) to be destabilized in multiple rat organs and HeLa cell line protein extracts upon treatment with Metformin (Fig. 7E). HexB is a lysosomal protein that hydrolyzes GM2 gangliosides and the specific activity of the HexB can be assayed using a fluorometric reporter[39]. However, we did not observe a large reduction in recombinant HexB activity caused by 1 mM Metformin when the assay was performed in neutral or acidic buffer to mimic the acidic pH of lysosomes[40] (Fig. 7F). Although reduction in activity was observed when using very high Metformin concentration at neutral pH (Fig. 7F), it can be hard to determine whether this reduction in activity was due to specific activity of Metformin. The assay was also performed on HeLa protein extracts and a similar trend was observed (Supplementary Fig. 7).

## Discussion

In this study, we present a comprehensive workflow for high-throughput TPP-PISA based on LFQ-DIA analysis, offering a robust framework for screening extensive compound libraries using the described methodology. Specifically, we introduced a specialized bioinformatics pipeline tailored for the analysis of large-scale compound screens in TPP-PISA experiments. This unique workflow harnesses datasets generated from expansive TPP-PISA-style experiments and simultaneously considers compound structural similarities. It enables the screening of a substantial number of drugs while utilizing the entire dataset as an internal control for each individual drug, thus providing enhanced statistical confidence in identifying potential drug targets. In an era of advancing mass spectrometry-based proteomics with ever-increasing sensitivity and throughput capabilities[41], the experimental workflows and bioinformatics pipelines presented herein assume a pivotal role. They could prove to be instrumental in handling the screening and analysis of substantial compound libraries, potentially comprising hundreds to thousands of unique small molecules.

Our study underscores the advantages of leveraging diverse proteome sources and the suitability of utilizing organs from model organisms such as Wistar rats. While human cell lines are invaluable for target engagement experiments due to their wide availability, comprehensively covering the proteome across different cell types and contexts necessitates the cultivation of numerous cell lines in large quantities. Furthermore, it could be difficult to discern whether the proteomes of cell lines differ sufficiently enough to cover different proteome contexts. In contrast, animal models present a compelling alternative, as the inherent differences in various organs are mirrored in their proteome compositions. This was exemplified by the

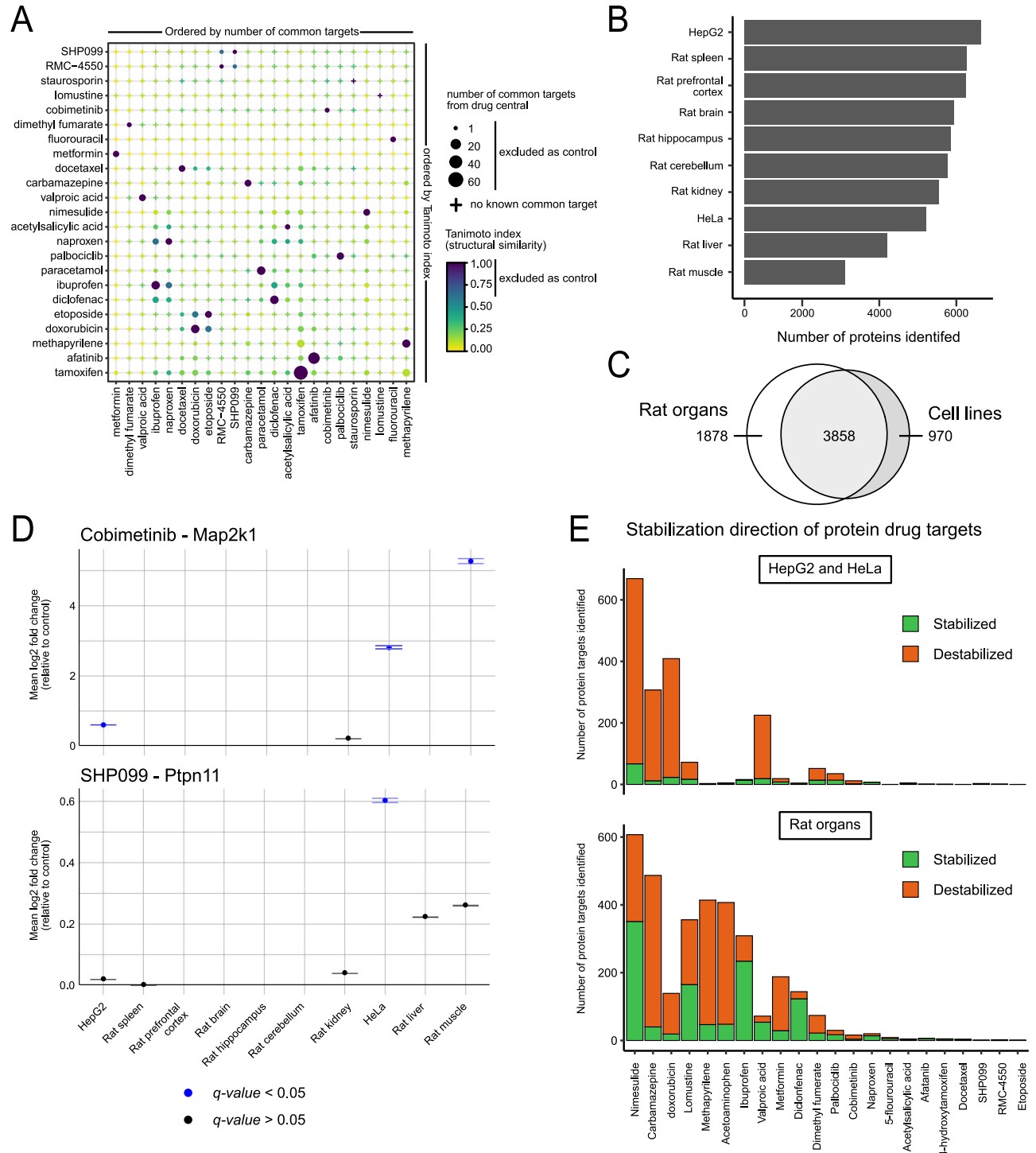

**Fig. 5 | Global analysis of drug targets and their effect on the proteome. A** Drug similarity plot of the number of common protein targets based on DrugCentral and structural similarity (Tanimoto index) for all the pairs of drugs used in this study. The points are ordered by hierarchical clustering based on their number of common targets (horizontal) and their structure similarity (vertical). **B** Number of proteins identified from TPP-PISA experiments for all the biological sources. **C** Overlap of the identified proteins from rat organs and human cell lines based on the gene names in the protein groups. **D** Temperature stabilization of Map2k1 and

Ptpn11 upon incubation with Cobimetinib and SHP099 respectively, after TPP-PISA analysis in different rat organ and human cell line protein extracts. Points represent the mean log-transformed fold change between the treated and control conditions, and the standard deviation is represented by error bars. Four replicates ($n = 4$) for each condition and comparison were used to derive the standard deviation. **E** Total number of significantly stabilized/destabilized protein groups identified upon heating in the presence of the different drugs tested in the study for human cell lines and rat organ extracts (top and bottom panel, respectively).

differential temperature stabilization direction of some kinases upon Staurosporine in different rat tissues. This could indicate more complex behavior of kinases due to the organ-specific kinase abundance or that of the surrounding proteome context. Consequently, rat tissues

offer a substantial advantage in screening different molecules to encompass a range of proteome backgrounds and the contexts in which these proteins are expressed. They also provide a surplus of protein extract material in comparison to cell lines. Obtaining a similar

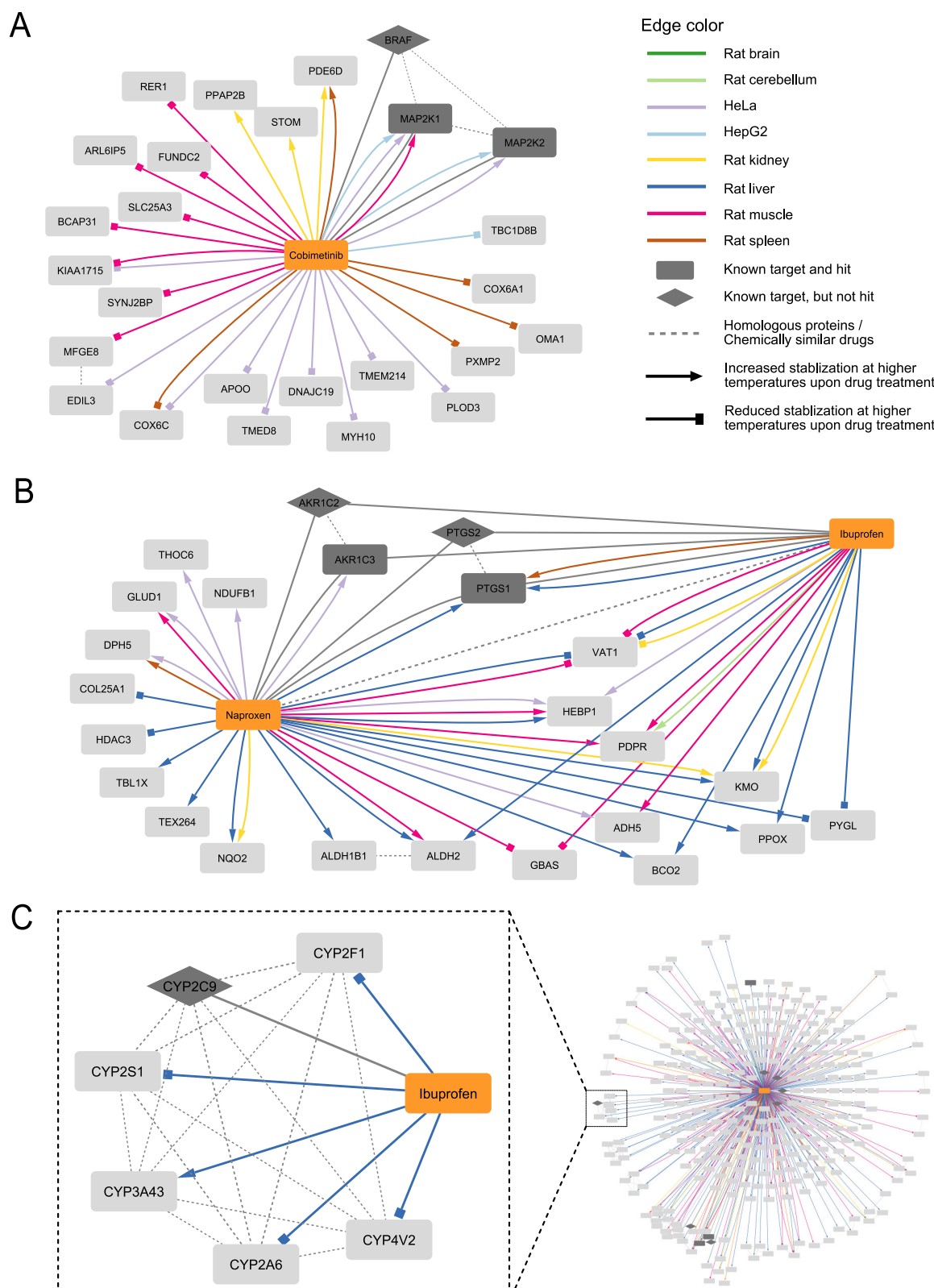

quantity of material typically required for TPP-PISA type experiments from traditional cultured human cell lines could prove impractical.

We used our workflow to identify Pirin as a target for the common nonsteroidal anti-inflammatory drug Ibuprofen. Pirin is a broadly conserved protein that is found in different domains of life including plants, fungi, and bacteria[42]. In humans, Pirin is ubiquitously expressed in various human tissues and organs[43]. Limited research into Pirin

proteins has implicated Pirin as a transcriptional regulator that plays an important role in the immune response and oxidative stress[44–47]. However, subcellular proteomics studies have primarily found the protein to be exclusively located in the cytosol[48,49]. Dysregulation of Pirin has been reported to play a role in various tumorigenesis pathways and cancers such as melanoma and colorectal cancer for instance[50,51]. Corollary, Ibuprofen which is a nonsteroidal anti-

**Fig. 6 | Drug target network visualization.** Networks consist of drugs (orange rectangles), known drug targets (dark gray diamonds), and significant targets from our study (light gray rectangles), which can be dark gray if they are also known targets. Solid non-gray edges represent a significant association between a tested drug and a target protein and are colored based on the rat organ extract or human cell line, in which they were identified. The arrow heads indicate the direction of temperature stabilization (delta for stabilizing and square for destabilizing). Edges between drugs and known targets from DrugCentral are shown as solid gray lines, while protein-protein homology and drug-drug similarity edges are represented as gray dashed lines. This legend applies to all subsequent network figures. **A** Protein target network of Cobimetinib showing all its protein targets identified across different extracts and the nature of the target (i.e. protein stabilized or destabilized). **B** Protein targets of Naproxen and common protein targets with Ibuprofen are shown in this network to demonstrate the analysis of common targets and the drug crosstalk. **C** Network of selected cytochrome P450 enzyme targets of Ibuprofen that share sequence homology with a known target of Ibuprofen according to DrugCentral.

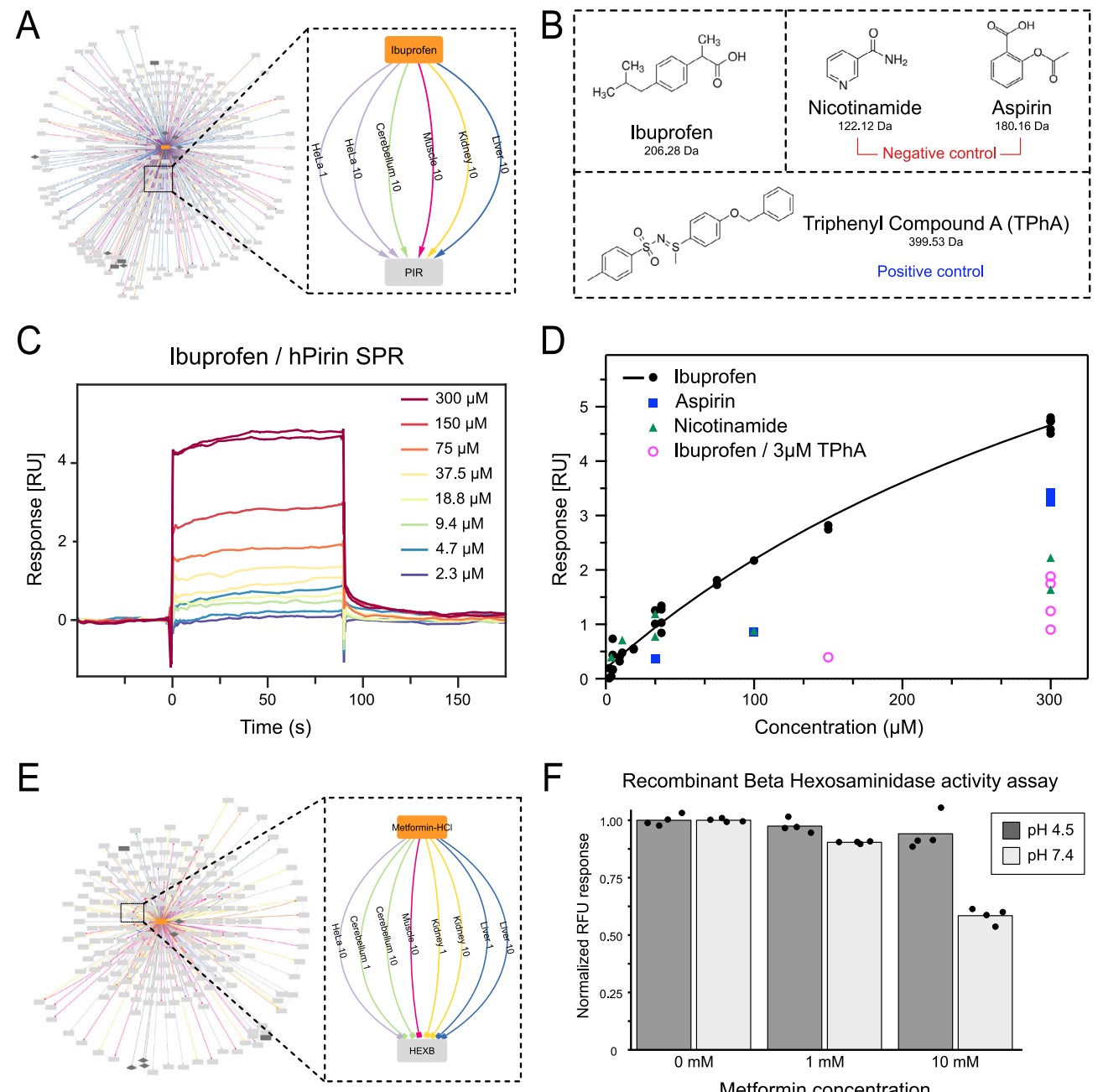

**Fig. 7 | Validation of Pirin as an Ibuprofen target. A** Protein target network of Ibuprofen highlighting Pirin as a target (via temperature stabilization) in multiple biological protein extracts. **B** Compound structures of Ibuprofen, the negative controls Nicotinamide and Aspirin as well as the positive control Triphenyl Compound A (TPhA) used for surface plasmon resonance (SPR) on recombinant human Pirin (hPirin). **C** SPR sensorgram of Ibuprofen at different concentrations on immobilized hPirin showing concentration-dependent increase in SPR response. **D** SPR dose-responses for Ibuprofen, negative controls, and Ibuprofen after TPhA pretreatment on hPirin. **E** Protein target network of Metformin highlighting HexB as a temperature destabilized target in multiple biological protein extracts. **F** Normalized Recombinant HexB (20 ng) activity using a fluorescent assay with 1 mM and 10 mM metformin at pH 4.5 and pH 7.4 is presented. Relative fluorescence unit was normalized to the mean of 0 mM value for pH 4.5 and pH 7.4 respectively. Individual replicates are plotted ($n = 4$). The legend of the networks can be found in Fig. 6.

inflammatory drug (NSAID), has been reported to potentially reduce risk significantly for colorectal cancer[52,53]. Nonetheless, the results demonstrate how this workflow can be employed to find additional targets for different drugs, even those with relatively weak interaction as is the case with Ibuprofen and Pirin. The weak interaction of Ibuprofen with Pirin is not unexpected given the small size and molecular weight of Ibuprofen. However, the results underscore the benefits of screening small compounds and fragments using this technique to aid drug design and test new hypotheses for their mechanism of action, which are often not well understood.

One of the surprising findings in our analysis was that, on average, a greater number of proteins exhibited a temperature destabilization effect rather than stabilization when treated with various drugs in different protein extracts (Fig. 5E). Thermal shift assays are grounded on the assumption that specific small molecule interactions with the target protein, like van der Waals forces or hydrogen bonding, reduce the Gibbs free energy, resulting in increased thermal stabilization. While it is theoretically possible that a protein-drug interaction could lead to conformational changes exposing potential hydrophobic patches that decrease thermal protein stabilization, this phenomenon is seldom observed[54–56]. Previous research has shown that changes in protein-protein interactions and large protein complexes can influence the thermal stability of component proteins[57,58]. Thus, the temperature destabilization of proteins upon drug treatment could predominantly result from downstream (secondary or tertiary) effects of the drug rather than direct engagement with the drug. The findings from this study would suggest that the effect of small molecules on the proteome could be through perturbation of protein-protein complexes or interactions, rather than through the direct targeting of protein binding pocket or inhibition. Consequently, the results of our analysis suggest that a significant portion of the physiological effects of small molecule compounds may arise from downstream or in-direct protein perturbations that may require us to re-evaluate the question of what constitutes a drug "target". This distinction may prove challenging to discern but could nonetheless have profound implications for drug discovery and development. This was exemplified by the lysosomal protein HexB, which exhibited destabilization in numerous organ and cell extracts following metformin treatment, despite no direct impact on its enzymatic activity at Metformin concentration below 1 mM. It is also possible that the effect observed between HexB and Metformin could be an artifact as no effect was observed on HexB activity in acidic pH buffer which was similar to what is expected in lysosomes.

One of the limitations in methods and variations of TPP, as employed in this study, is its bias against proteins that are difficult to monitor with the strategy presented here, such as membrane proteins. Although substantial efforts have been made to broaden membrane protein coverage, there remain inherent constraints within the technique itself that hinder complete coverage[15]. Furthermore, in our pursuit of increased throughput, our workflow entailed the use of native cellular protein extracts rather than conducting experiments within live cells or in vivo. While this approach aims to preserve the native conformation of proteins and protein complexes, it does result in the disruption of cellular compartments, potentially introducing inaccuracies into target identification which might explain why we observed HexB as a metformin target. The native cellular environment of proteins could explain why a particular protein was observed as a target in one rat organ or cell line protein extract but not in another, despite being detected. This could be caused by lower abundance and lower reproducibility in certain organs such that protein targets do not pass statistical threshold criteria. It is also possible that the presence or absence of interaction partners of the target protein might be required for the observed action of a drug on a protein, which may occur in some rat organs or cell lines in a context-dependent

manner. Additionally, there is the likelihood that an unknown number of direct or indirect target proteins of small molecule drugs simply fall below the detection threshold of this workflow due to the limited temperature resolution. It is also crucial to acknowledge that the effect of additives in the protein extraction and solubilization buffer, such as protease and phosphatase inhibitors, could prevent identification of proteases and phosphatases as drug targets in TPP and PISA assays. One limitation of PISA is that this method is not able to discriminate between abundance and thermal stability changes upon compound treatment when applying the workflow to living cells. This could be addressed through proteome measurement prior to the TPP temperature incubation, which could be used to normalize the abundance changes observed in the PISA measurement. Furthermore, we did not assess the impact of post-translational modifications on the stability of the protein which could further complicate the analysis. Unsurprisingly, DIA analysis resulted in a larger number of protein identifications when compared directly with DDA-based TMT TPP analysis (Fig. S2). However, peptide fractionation can lead to higher proteome coverage and sensitivity of DDA-based TPP resulting in the identification of more drug targets[20]. Despite these limitations, we aimed to strike a balance between throughput, sensitivity, and proteome depth with our workflow. As such, we anticipate future optimizations in thermal proteome profiling-based methods that will complement other techniques in drug development and target identification.

## Methods

### Ethical statement

Animal experiments were conducted following the guidelines of the Danish Animal Experimentation Inspectorate (license number: 2016-15-0201-00934) and approved by the local institutional animal care unit.

### Cell lines protein extraction

HeLa and HepG2 adherent cell lines were purchased from ATCC (Manassas, Virginia, USA). Both cell lines were cultured in DMEM (Gibco) containing 100 U/mL penicillin (Invitrogen), and 100 mg/mL streptomycin (Invitrogen) and supplemented with 10% bovine calf serum (Gibco) at 37 °C in a humidified incubator with 5% $CO_2$. Cells were grown to 80% confluency and washed with phosphate-buffered saline (PBS) 5 times to ensure the removal of all media and serum components. Cells were scraped and collected with a cold PBS buffer. Cells were then collected by centrifugation at 400 × g for 3 minutes and the supernatant was removed. Cold PBS buffer was added and the cells were gently resuspended again followed by another centrifugation and collection step. This was repeated two more times for a total of 4 washes. After the last centrifugation step, the supernatant was removed, and a precooled native protein extraction buffer added to the cells and gently resuspended. The extraction buffer composition was: 30 mM HEPES (Sigma-Aldrich) pH 7.4; 0.2% NP-40, 100 mM NaCl, 2 mM $MgCl_2$, 2.7 mM KCl, 10 mM $Na_2HPO_4$, and 1.8 mM $KH_2PO_4$1. The extraction buffer was additionally supplemented with 5 mM β-glycerophosphate, 1 mM sodium orthovanadate, 5 mM sodium fluoride, 1 mM phenylmethylsulfonyl fluoride and EDTA free protease inhibitor cocktail (Roche). After the addition of protein extraction buffer, the sample was flash frozen with liquid nitrogen and stored at −80 °C or thawed at room temperature followed by two additional freeze-thaw cycles in a similar manner. Benzonase was then added to the sample and rotated in a cold room (4 °C) for 1 hour. The sample was centrifuged at 2000 × g for 10 minutes at 4 °C and the supernatant was collected in a new tube. The collected supernatant was passed through a 0.45 μm filter after which the protein concentration was determined using a Bradford assay and aliquoted prior to storage at −80 °C. The resulting material was then utilized for TPP and TPP-PISA assays.

## Rat organ extraction

Adult male Wistar rats (200-220 g at arrival; Charles River, Sulzfeld, Germany) were anesthetized with pentobarbital (200 mg/ml) mixed with lidocaine (20 mg/ml) and transcardially perfused with ice-cold 0.9% saline supplemented with 10 U/ml heparin (Heparin LEO, LEO pharma, Denmark). Additional perfusion was performed with ice-cold 0.9% saline solution supplemented with 5 mM β-glycerophosphate, 1 mM sodium orthovanadate, 5 mM sodium fluoride, 1 mM phenylmethylsulfonyl fluoride and solubilized EDTA-free protease inhibitor cocktail (Roche). Dissected organs were immediately snap-frozen in liquid nitrogen and stored at −80 °C until further processing. Frozen rat organs were either stored at −80 °C or further processed for native protein extraction. Organs were transferred to 15 ml conical tubes, and cold native protein extraction buffer as described above was added to the organs with the exception that 0.4% NP-40 was used instead of 0.2%. Organs were pulverized using a Ultra Turbax blender (IKA, Staufen, Germany) until solubilization of the organ material was sufficiently achieved. Sample was kept on ice and centrifuged at 4 °C for 20 minutes at 5000 x g. Supernatant was transferred to another conical tube and benzonase was added to the sample and rotated in a cold room (4 °C) for 1 hour. Following benzonase treatment, the supernatant was passed through a 0.45 μm filter after which the protein concentration was determined using a Bradford assay and aliquoted prior to storage at −80 °C and the resulting material was utilized for TPP-PISA assays.

## TPP experiment

In the Thermal Proteome Profiling (TPP) experiment, samples were treated with Staurosporine or DMSO for 10 minutes and then divided into 10 different temperature conditions (37, 41, 44, 47, 50, 53, 56, 59, 63, 67 °C) for 3 minutes. After temperature treatment, the samples underwent centrifugation for 30 minutes at 20,000 × g, and the resulting supernatant was carefully transferred to new tubes. For mass spectrometry analysis, the Protein Aggregation Capture protocol (PAC) was applied[30]. Briefly, proteins were aggregated with the addition of acetonitrile (to a final concentration of 70%). ReSyn Bioscience (Gauteng, South Africa) magnetic hydroxyl beads were added onto which proteins were aggregated and separated by magnet. Aggregated proteins were washed with 100% Acetonitrile and 70% Ethanol followed by the addition of digestion buffer (50 mM HEPES, pH 8.5). This was followed by sequential digestion with Lys-C (Wako-Fujifilm, Tokyo, Japan) at a 1:100 (protease to protein ratio) at 37 °C for 2 hours, followed by an overnight digestion with Trypsin (Promega, Madison, Wisconsin, USA) at a 1:100 ratio at 37 °C. After the overnight digestion, acetonitrile was added to achieve a final concentration of 50% and the beads were magnetically removed. Tandem Mass Tags (TMT) labeling was then carried out as described in the subsequent section.

## TPP-PISA

For TPP-PISA analysis, the procedure involves treating samples with a drug at 37 °C in a 96-well plate. Subsequently, each sample is evenly divided into three wells of the same volume. These divided wells are then subjected to heating at different temperature blocks (50, 53, and 56 °C, respectively) for 4 minutes using an Applied Biosystems Veriti Thermo Cycler (Thermo Fisher Scientific) followed by a ramp down to cool the plate to 4 °C. To consolidate the samples, the contents from the divided wells were pooled back into the original well.

## Insoluble protein removal

To eliminate insoluble proteins, two methods are employed. In the first method, centrifugation is performed for 30 minutes at 20,000 × g. Alternatively, for the optimized protocol, the insoluble fraction is removed by transferring TPP-PISA treated samples to a hydrophilic polytetrafluoroethylene 96-well filter plate with a 0.45 μm membrane pore size (MSRLN0410, Millipore, Burlington, Massachusetts, USA).

The filter plate is pre-wetted with 100 μl of PBS and centrifuged for 2 minutes at 500 × g. TPP-PISA samples are added, centrifuged for 5 minutes at 500 × g and the flowthrough is collected in a new plate.

## Sample preparation and digestion

Following the removal of insoluble proteins, 4% SDS is added to the filtered samples. Reduction and alkylation are performed simultaneously using 5 mM tris(2-carboxyethyl) phosphine and 5.5 mM chloroacetamide respectively for 30 minutes. Subsequently, samples are prepared using the protein aggregation protocol (PAC) in the same well plate. Sequential digestion is carried out with Lys-C (Wako-Fujifilm, Tokyo, Japan) at a 1:200 ratio (protease to protein) at 37 °C for 2 hours, followed by overnight digestion with Trypsin (Promega, Madison, Wisconsin, USA) at a 1:100 ratio at 37 °C. Post-overnight digestion, samples are acidified with TFA (1% final). Magnetic beads are removed by magnet and the supernatants are transferred to a new plate. For samples to be analyzed using the Evosep One system (Evosep, Odense, Denmark), the supernatants are desalted and concentrated on Evotips (Evosep, Odense, Denmark) for storage and analysis.

## TPP TMT labeling

Tandem mass tags 10-plex reagents (Thermo Fisher Scientific, San Jose, USA) were utilized to label the resulting peptide samples with the temperature points representing different mass tags. The labeling process involved allowing the reaction to proceed for 1 hour at room temperature. The reaction was quenched by adding 1% hydroxylamine for 15 minutes, and the mixture was acidified with trifluoroacetic acid (1% final concentration). Samples labeled with different TMT-plexes were pooled, and acetonitrile was evaporated using a SpeedVac (Eppendorf, Germany) operating at 45 °C. Peptides were reconstituted in 0.1% TFA, and subsequent steps included solid-phase extraction using C18 Sep-Paks (Waters Corporation, Milford, MA, USA), washing with 0.1% formic acid, and elution with a buffer containing 40% acetonitrile and 0.1% formic acid. Acetonitrile from the eluted samples was evaporated using the SpeedVac at 45 °C, and peptide concentration was determined using a NanoDrop 2000 spectrometer (Thermo Fisher Scientific, Waltham, MA, USA).

## Mass spectrometry analysis

All samples were analyzed using a Q-Exactive HF-X or Orbitrap Exploris 480 mass spectrometer (Thermo Fisher Scientific) operating in positive ion mode coupled to an EvoSep One (EvoSep, Odense, Denmark) or EASY-nLC 1200 (Thermo Fisher Scientific) liquid chromatography system. Samples injected with the EASY-nLC 1200 system were separated on a in house emitter-column packed with 1.9 μm $C_{18}$ beads (Dr. Maisch GmbH, Entringen, Germany). Peptides were loaded and washed using a 0.1% formic acid, 5% acetonitrile buffer in Buffer A, and the peptides were eluted using Buffer B composed of 0.1% formic acid (FA) and 80% acetonitrile. For TPP and TPP-PISA samples with TMT labeled peptides, a 220-minute gradient going from 5% to 25% Buffer B was used, followed by a 20-minute increase to 40% Buffer B. The column was subsequently washed by increasing Buffer B to 80% in 5 minutes, where it was held for an additional 5 minutes. This was followed by a decrease back to 5% Buffer B in 5 minutes, where it was held for an additional 5 minutes for column equilibration. For samples analyzed with a coupled EvoSep One LC system, peptides were separated on a 15 cm $C_{18}$ analytical LC column (0.15 × 150 mm, 1.9 μm beads, EV-1106) utilizing the standardized 30 samples-per-day (SPD) method, which is a 44-minute active LC gradient with a total run time of 48 minutes. The analytical column was equilibrated at room temperature and operated at a flow rate of 500 nL/min. The gradient was obtained by flow injection of Evosep solvent A ($H_2O$/0.1% FA) and solvent B (acetonitrile with 0.1% FA) and peptides eluted by a linear gradient up to 35% solvent B. For TMT labeled samples, a data-dependent acquisition (DDA) top 12

method with 120,000 and 45,000 MS1 and MS2 resolution was used, respectively. For label free samples analyzed using DDA, a Top 20 method was used with 120,000 MS1 resolution and 15,000 MS2 resolution. For data-independent analysis (DIA) on the Exploris 480 Orbitrap mass spectrometer, MS1 resolution of 120,000 and MS2 resolution of 15,000 were chosen for the DIA scans. Isolation window size of 13.7 mass-to-charge (m/z) was used with 1 m/z charge overlap between the isolation bins. DIA mass range of 361-1033 m/z was used for a total of 49 scan events and HCD normalized collision energy of 27% was utilized for fragmentation.

## Mass spectrometry raw file analysis

DDA raw files generated from mass spectrometry analysis were analyzed by MaxQuant version 1.5.7.0 software with Andromeda search engine[59]. Raw data was searched against the human UniProt FASTA database (UP000005640) with "Trypsin/P" used for protease specificity. Contaminant FASTA from MaxQuant was utilized to search and filter for common contaminant proteins. Additionally, 2 missed cleavages were allowed, carbamidomethylation of cysteine was set as a fixed modification while methionine oxidation and protein N-terminal acetylation were set as variable modifications. Mass tolerance for MS1 in the first search was set to 20 parts per million (ppm) followed by 4.5 ppm after calibration. Mass tolerance for MS2 was set to 20 ppm. For TMT analysis, "Reporter ion MS2" was used as a search parameter and 10-plex TMT was utilized for reporter groups. 1% false discovery rate (0.01) was utilized at the peptide spectral match and protein level.

DIA raw files were analyzed using DIA-NN version 1.81[28]. Library free search was accomplished using an in silico generated spectral library within DIA-NN. Trypsin/P was used as a protease, 1 missed cleavage was allowed, carbamidomethylation of cysteine was set as fixed modification, precursor charge state 2-5 was used, and the precursor FDR (%) was set to 1.0. For human cell line analysis, the UniProt FASTA database (UP000005640) was used. For searching data generated from rat organ extract experiments, the protein sequences from *Rattus norvegicus* were retrieved from BioMarts the 28th of March 2021. A contaminant FASTA protein database from MaxQuant was also used alongside both human and rat sequences for in silico generation of spectral libraries. Match between runs was selected and double-pass mode setting was used as a neural network classifier. Robust LC (high accuracy) and RT-dependent settings were used for quantification strategy and cross-run normalization respectively.

## Proteomics data analysis

For TPP-PISA data (Fig. 2), the corrected TMT reporter intensities were retrieved from the "proteingroups.txt" file generated from MaxQuant analysis. Data were log2-transformed and normalized using quantile-based normalization after removing reverse sequences, potential contaminants, and protein groups with no quantified proteotypic peptides. "0" were replaced by NAs. For TPP analysis, corrected reporter intensities of TPP raw files analyzed by MaxQuant were used. Reporter intensities from different temperatures for all TPP samples were normalized to the TMT channel corresponding to 37 °C (TMT-126 for all samples) to convert the intensity values to ratios. The converted ratios were used for melting point analysis of identified proteins using the TPP package[13]. For comparison between TPP and TPP-PISA, we performed two-sided unpaired t-tests (equal variance) followed by Storey correction for multiple testing on the protein group melting temperatures and relative quantities. For the figures, protein groups were labeled kinase according to the information retrieved from kinhub.org/kinases.html (01/12/2020).

For the other experiments, DIA data were analyzed using DIA-NN, and we utilized the search outputs "pg_matrix" for downstream analysis. Contaminant proteins were removed, and the protein group intensity values were log2 transformed before re-alignment of median signal across runs. Runs with more than 40% of missing values were excluded from the analysis. Protein group signals from replicate MS runs of the same condition (drug treatment) were averaged. When independent MS-search outputs from the same species were combined (for Euler diagrams for example), we mapped the exact same protein group identifiers as provided by DIA-NN. When comparing species, rat protein identifiers from Ensembl were matched to their gene names as well as to human homologs using the Ensembl Compara ortholog mapping[60] provided by BioMart from the FASTA rat identifiers (download 2nd of August 2021).

## Surface plasmon resonance

All SPR experiments were performed at 25 °C on a Biacore T200 instrument equipped with a Series S CM5 sensor chip (Cytiva, Uppsala, Sweden). SPR running buffers and amine-coupling reagents (1-ethly-3-(3-dimethylaminoproply)carbodiimide hydrochloride (EDC), N-hydroxysuccinimide (NHS), and ethanolamine hydrochloride-NaOH pH 8.5 were purchased from Cytiva. DMSO was from Sigma Aldrich (34943). 1x PBS-P, 2% DMSO (11.9 mM $NaH_2PO_4$-$Na_2HPO_4$ pH 7.4, 137 mM NaCl, 2.7 mM KCl, 0.005% (v/v) surfactant P20, 2% DMSO) was used as running buffer. Recombinant human pirin (Abcam, ab123170) was buffer exchanged to PBS and immobilized at 25 µg/ml to ~ 10,000 RUs on flow cell 2 (Fc2) of the CM5 chip using standard EDC/NHS coupling chemistry and following the manufacturer's protocol. The flow cell 1 (Fc1) was as well EDC/NHS treated and used as a reference cell for subtraction of systematic instrumental drift. Triphenyl Compound A (TPhA) was purchased from MedChemExpress (HY-14454 catalog number). Nicotinamide was purchased from Sigma Aldrich (72340). Ibuprofen and Aspirin were purchased from Merck Sigma-Aldrich. Concentrations of stock compounds in DMSO were 10 mM (TPhA) or 135 mM (Ibuprofen, Aspirin, Nicotinamide). For SPR experiments the compounds were diluted first in DMSO free SPR running buffer to 200 µM (TPhA) or 2700 µM (Ibuprofen, Aspirin and Nicotinamide) and then further diluted with SPR running buffer containing 2% DMSO to 9000 nM (TphA) or 300 µM (Ibuprofen, Aspirin, Nicotinamide) highest concentrations. Three- or two-fold serial dilutions were subsequently prepared in running buffer, and each concentration series was injected sequentially for 90 s from lowest to highest concentration over flow cells 1, 2 at a flow rate of 60 µl/min and 120 s (TPhA, Ibuprofen, Aspirin, Nicotinamide) dissociation times. Ibuprofen was tested as well in a SPR running buffer supplemented with 3 µM TPhA to test for the possibility of competitive binding between TPhA and Ibuprofen. Eight solutions with increasing concentrations of DMSO (1.5−2.8%) were prepared to build a solvent correction curve that accounts for variations in bulk response due to small percentage variation in DMSO concentration across samples. Data processing and analysis was done using the BiaEvaluation software (v. 3.2.1, Cytiva, Uppsala, Sweden). The raw sensorgrams were solvent corrected, double referenced (referring to the subtraction of the data over the reference surface and the average of the buffer injections from the binding responses), and the equilibrium dissociation constant, $K_D$, was determined using a steady state model (for TPhA, Ibuprofen).

## Beta hexosaminidase (HexB) activity assay

HexB assay was conducted according to manufacturer instructions (Cell Biolabs Inc, San Diego, California, USA). Briefly, 20 ng of recombinant human HexB was incubated with substrate for 30 minutes at 37 °C in a 96-well black plate followed by reaction quenching using a neutralization buffer. Tecan (Männedorf, Switzerland) Spark microplate spectrometer was used for fluorescent measurements with an excitation and emission wavelengths of 365 and 450 nm respectively. Relative fluorescent units (RFU) were used for reported data analysis. For Fig. 7F, all points were normalized to the mean RFU of 0 mM Metformin for pH 4.5 and pH 7.4 respectively.

## Statistical analysis of the TPP experiments

Data from the same experiment were analyzed together with two-sided unpaired t-tests (equal variance). For the Pilot experiments, we compared the 4 drug-treated replicates with the corresponding DMSO controls of the experiment. In the screen data (human cells and rat organs), we compared the 4 replicates of a given drug and concentration to all other conditions excluding: i) the same drug at different concentrations; ii) drugs with a Tanimoto index ≥ 0.6; iii) drugs with minimum one common target. $p$-values were adjusted for multiple testing using Storey ("qvalue" package v2.22.0 in R). The combined outputs of each statistical test are available at the Zenodo data repository (https://doi.org/10.5281/zenodo.13141610), drugs excluded from the controls are in Supplementary Data 4. For the screen, we considered significantly regulated protein groups with a $q$-value ≤ 0.05 and an absolute log2-transformed fold change ≥ 0.5.

## Drug similarity

Known drug target interactions and drug structures (smiles) were retrieved from DrugCentral[35] on drugcentral.org the 13/01/2021 and 03/08/2021, respectively. The smiles of SHP099 and RMC-4550 were not available in DrugCentral and were retrieved from biomol.com (06/09/2021). The Tanimoto index reflecting pairwise drug similarity was calculated with Open Babel v3.1.0[61]. The outputs of this analysis are in Supplementary Data 4.

## Protein–drug network construction and visualization

A protein–drug network was constructed from the data such that each edge corresponds to a significant association between a tested drug and a target protein ($q$-value ≤ 0.05 and an absolute log2-transformed fold change ≥ 0.5, see the paragraph "Statistical analysis of the TPP experiments" for more details), identified in at least one of the rat organ extracts or human cell lines. Some targets were identified in multiple experiments resulting in several parallel edges between this target and the drug. To combine the rat and human data in one network visualization, rat proteins were mapped to their human orthologs using BioMart as described for the proteomics data analysis and the first identifier in each group was used as a representative. The network was further expanded by drug–drug similarity edges based on a Tanimoto index > 0.6 as well as protein–protein similarity edges based on their homology scores retrieved from STRING v11.5 for human without further cutoff[62]. Known drug–target associations from DrugCentral were retrieved on 13/01/2021 and added as edges to the network for the tested drugs. Network visualizations were created using Cytoscape[63] and available at https://doi.org/10.5281/zenodo.13141610.

## Reporting summary

Further information on research design is available in the Nature Portfolio Reporting Summary linked to this article.

# Data availability

The mass spectrometry data generated in this study have been deposited in the ProteomeXchange Consortium via the PRIDE[64] partner repository with the dataset identifier PXD050784. Source data are provided with this paper.

# Code availability

The data and the scripts associated with this study are available at https://doi.org/10.5281/zenodo.13141610 under a BSD 2-Clause "Simplified" License.

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

## Acknowledgements

Work at The Novo Nordisk Foundation Center for Protein Research (CPR) is funded in part by a donation from the Novo Nordisk Foundation (NNF14CC0001). The proteomics technology developments applied were funded by the European Union's Horizon 2020 research and innovation program under grant agreement EPIC-XS-823839 and the Independent Research Fund Denmark Medical Sciences Instrument DFF-Research Project 1 (2034-00445 A). We additionally thank Stanislava Pankratova for assistance with the Wistar rats.

## Author contributions

T.S.B. and J.V.O. proposed the study. T.S.B., J.V.O., M.L.P., and L.J.J. developed the scientific strategy. T.S.B. designed and carried out the experimental, proteomic strategy and analyzed all proteomics-related samples. M.L.P. developed the bioinformatics and statistical strategy for the analysis and interpretation of the data. N.T.D. performed drug-protein network analysis and protein ortholog mapping across species. B.L.M. performed the SPR analysis.

## Competing interests

The authors declare no competing interests.
