## [Transparent Peer Review file · Nature Communications]

Streamlined analysis of drug targets by proteome integral solubility alteration indicates organ-specific engagement

Corresponding Author: Professor Jesper Olsen

Version 0:

Reviewer comments:

Reviewer #1

(Remarks to the Author)

In this manuscript, Batth et al present a streamlined workflow for thermal proteome profiling to do it 1. In larger throughput and 2. On rat organs.

Specifically, they changed current TPP and PISA protocols to use only specific temperature points in a PISA approach, utilise filter plates instead of ultracentrifugation, and employ DIA rather than TMT and DDA. The workflow is now more high-throughput compatible and appears to provide more details than the standard TPP.

The authors then apply the workflow to a system in which rat organ extracts were treated with staurosporine.

They continue then using 22 compounds in these rat extracts and human cell lines. They follow up on a new potential protein binder of ibuprofen. This interaction is validated by SPR, but weak.

This is a technically very well-done work. It is well written. It will be of interest to a wider range of proteomics and drug discovery groups utilising TPP or similar approaches. The new finding for some of the drugs may be of interest to a wider community.

I have no major comments.

Minor comments:

Figure legend 1 CETSA and not CESTA

I am pretty sure that xG should be xg as G is the gravitational constant and g the acceleration through gravity.

Are the authors sure that utilising 5mM β glycerophosphate, 1mM sodium orthovanadate, 5mM sodium fluoride, 1mM phenylmethylsulfonyl fluoride and solubilized EDTA free protease inhibitor cocktail (Roche) in the perfusion cocktail would not affect the thermal stability of some proteins?

Embarrassingly, I would like you to consider citing our paper when it comes to the use of DIA in TPP. I think it would be appropriate. <https://pubmed.ncbi.nlm.nih.gov/37439223/>

Please provide the Kd for ibuprofen binding to Pirin in the text.

Reviewer #2

(Remarks to the Author)

Batth et al. optimized an existing PISA method for high-throughput screening of drug targets and validated its utility in various rat organs. Compared to the traditional TPP method, PISA uses fewer temperature points to capture drug-induced changes in melting curves. Overall, although not technically superior, this is a potentially broadly applicable approach that could benefit the field of drug discovery and biology.

Major issues:

The title of the paper should directly state that the reported method was optimized/developed based on the TPP-PISA approach.

What is the false discovery rate and true discovery rate when using DIA-TPP-PISA compared to TMT-TPP and TMT-TPP-PISA? The comparison of protein ID and throughput alone (Fig. S2) is not sufficient for method benchmarking.

In addition, only staurosporine is used to validate the soundness of the method. Can the authors use at least two model drugs rather than just one compound to test the ability of the method to identify true drug targets?

The authors reported the DIA-TPP-PISA method using filter plates rather than centrifugal precipitation. This greatly simplifies the TPP procedure. However, it would be useful to explain why the PTFE filter with a pore size of 0.45 µm was used. Please characterise the proteins retained/eluted by the filters with different pore sizes.

Why does the stability of staurosporine target kinases change in different tissues? Is this change related to organ-specific expression levels of the kinases or other metrics? Please analyze and discuss the reasons for this.

Although the initial amount of protein is identical in each group, the amount of soluble protein from each sample after heating is not equivalent across temperature groups. How do the authors perform the 'cross-run normalization' correction when examining the LFQ DIA data? Does this normalisation step reduce the observed protein differences?

Minor points.

1. The authors only included methods such as DARTS, Lip-MS and CETSA. A more comprehensive review of the current target ID literature, such as Thermal Proteome Profiling (TPP), Solvent Induced Protein Precipitation (SIP) and Target-Responsive Accessibility Profiling (TRAP), should be included.

The section "Mass spectrometry analysis" lacks a description of the liquid phase gradient. Also, please add whether the LC gradient and instrumentation differ between DIA and TMT acquisitions.

In line 122, "10 µm staurosporine" should be corrected to "10 µM staurosporine".

Different yellow series lines in Figure 6 are difficult to discern.

Reviewer #3

(Remarks to the Author)

Bath et al. describe a Proteome Integral Stability Analysis workflow using a data-independent acquisition mass spectrometry-based proteomics protocol. By applying it to profile 22 compounds in different cell and tissue systems the authors find a previously undescribed target of the commonly used NSAID Ibuprofen. Many of the protocol adaptations that the authors describe are not new and should be referenced correctly. Moreover, while the combination of adaptations to the workflow does have advantages over existing workflows, as the authors rightfully point out, a careful comparison to existing approaches is needed to give potential future users a clear overview of what is best for their use case.

Major points:

- While disadvantages of TMT data-dependent acquisition such as limited multiplexing capability are discussed, advantages such as higher sensitivity are not mentioned. This point would become more evident when data independent acquisition (DIA) TPP/PISA data would be compared to a dataset derived with data-dependent acquisition (DDA) using TMT, e.g. using the Staurosporine dataset by the Savitski et al. (2014) paper which would make a nice benchmark to compare the new DIA approach with the previous DDA one. This could potentially be even extended by comparison to a newer dataset using TMT-18 by Zinn et al. (2021) Journal of Proteome Res.

- It would be also critical to use the latest analysis methods, e.g. nonparametric analysis of response curves (Childs et al. (2019) Mol. Cell. Proteomics) to compare TPP to PISA since the melting-point comparison does not make use of the full measurements in the TPP data and would likely show comparable or higher sensitivity in identifying kinase targets from the data.

- The replacement of the centrifugation step for protein aggregate removal and also the 96 well format (at least for TPP) is not new. It has been previously applied by Selkrig et al. (2021) Mol. Syst. Biol, please rephrase and reference accordingly.

- Abundance vs. thermal stability in PISA: the PISA assay cannot distinguish between changes in protein abundance (also solubility which can be modulated by some compounds, especially at higher concentrations) and thermal stability. In the lysate format this is not a problem, but if potential future users would like to apply the workflow to living cells this will be an issue, please add a cautionary statement to the discussion.

- The authors comment on one of the described targets of Ibuprofen: "CYP2C9 was not identified as a target in our analysis" is this because the protein was not measured or because it was not found to be changed in thermal stability. How are the numbers of identified protein in general comparing to DDA-based TPP datasets? Please extend discussion accordingly.

Minor points:

- The authors might want to acknowledge that Perrin et al. (2020) Nat. Biotechnology were the first to implement TPP in an in vivo and ex vivo setting of rodent organs.

Version 1:

Reviewer comments:

Reviewer #2

(Remarks to the Author)

I feel most of the issues have been fully addressed.

Reviewer #3

(Remarks to the Author)

The authors have addressed the large majority of my comments and I recommend the manuscript in its current form for publication with a minor addition: I would like the authors to add one more half sentence clarifying that the 'standard TPP' experiment they performed (p. 6) is not done using peptide fractionation (I guess to make run time comparable to DIA), which

is a crucial step in TMT-based TPP experiments to guarantee detection numbers and precision in peptide quantification and strongly influences the quality of the obtained data which in turn affects sensitivity and specificity in staurosporine target identification.

Point-by-point response to reviewers.

Manuscript: NCOMMS-24-26364-T

Batth et al.

We thank the reviewers for their constructive feedback. Please find our point-by-point response to each of the remarks. We have highlighted our response in **blue** color.

Reviewer #1 (Remarks to the Author)

In this manuscript, Batth et al present a streamlined workflow for thermal proteome profiling to do it 1. In larger throughput and 2. On rat organs.

Specifically, they changed current TPP and PISA protocols to use only specific temperature points in a PISA approach, utilise filter plates instead of ultracentrifugation, and employ DIA rather than TMT and DDA. The workflow is now more high-throughput compatible and appears to provide more details than the standard TPP.

The authors then apply the workflow to a system in which rat organ extracts were treated with staurosporine.

They continue then using 22 compounds in these rat extracts and human cell lines. They follow up on a new potential protein binder of ibuprofen. This interaction is validated by SPR, but weak.

This is a technically very well-done work. It is well written. It will be of interest to a wider range of proteomics and drug discovery groups utilising TPP or similar approaches. The new finding for some of the drugs may be of interest to a wider community.

I have no major comments.

We thank the reviewer for highlighting the high technical quality of our manuscript and its broad interest for the wider scientific community.

Minor comments:

Figure legend 1 CETSA and not CESTA

We thank the reviewer for identifying this typo. We have corrected the error in the figure. We have additionally corrected 2 more instances of this error in the main text as well.

I am pretty sure that xG should be xg as G is the gravitational constant and g the acceleration through gravity.

We thank the reviewer for identifying this error, and indeed lowercase xg (acceleration through gravity) should be used. We have corrected all instances of “xG” to “ x g” in the text and figure legends.

Are the authors sure that utilising 5mM β glycerophosphate, 1mM sodium orthovanadate, 5mM sodium fluoride, 1mM phenylmethylsulfonyl fluoride and solubilized EDTA free protease inhibitor cocktail (Roche) in the perfusion cocktail would not affect the thermal stability of some proteins?

The reviewer raises a very important point concerning the additives in the extraction buffer, and we agree that it is not something that has been discussed but warrants contemplation. The reviewer is correct that these compounds are likely to affect thermal stability of some proteins, furthermore, the addition of these compounds which primarily serve to block residual protease and phosphatase activity could prevent identification of those proteins as possible targets in TPP/PISA assays, especially if the compound is targeting similar binding site as these molecules. However, since these compounds are also in the control samples, we cannot ascertain the full extent of their effect. We have added the following remarks at the end of the discussion section :

“It is also crucial to acknowledge that the effect of additives in the protein extraction and solubilization buffer, such as protease and phosphatase inhibitors, could prevent identification of proteases and phosphatases as drug targets in TPP and PISA assays.”

Embarrassingly, I would like you to consider citing our paper when it comes to the use of DIA in TPP. I think it would be appropriate. <https://pubmed.ncbi.nlm.nih.gov/37439223/>

We thank the reviewer for suggesting this important reference and apologize for overlooking this paper. We have now added this reference to the text and modified the sentence in which it occurs on line 158:

“Although DIA analysis requires independent measurement of each sample it has been demonstrated to be compatible for melting curve fitting using the standard TPP workflow [George et. al reference]...”

Please provide the Kd for ibuprofen binding to Pirin in the text.

We have added the Kd in the text although it must be noted that as the curves do not reach saturation, the value given may not be accurate. We have added this in the text with the following statement:

“Using SPR, we observed a dose-dependent response of Ibuprofen binding to Pirin (Figure 7C) with a K_d of 546 μ M, although it can be difficult to accurately determine K_d as the curve did not reach saturation.”

Reviewer #2 (Remarks to the Author):

Batth et al. optimized an existing PISA method for high-throughput screening of drug targets and validated its utility in various rat organs. Compared to the traditional TPP method, PISA uses fewer temperature points to capture drug-induced changes in melting curves. Overall, although not technically superior, this is a potentially broadly applicable approach that could benefit the field of drug discovery and biology.

We thank the reviewer for the feedback and careful review of this manuscript.

Major issues:

The title of the paper should directly state that the reported method was optimized/developed based on the TPP-PISA approach.

We have changed the title of the paper to “Streamlined analysis of drug targets by proteome integral solubility alteration indicates organ-specific engagement”.

What is the false discovery rate and true discovery rate when using DIA-TPP-PISA compared to TMT-TPP and TMT-TPP-PISA?

We thank the reviewer for this important question. Aside from statistical tests and cutoffs to determine hits, we could approximate true hits based on kinase targets as Staurosporine is a broad kinase inhibitor. Figure 2E (TMT-TPP/TMT-TPP-PISA) and Figure 3D (DIA-TPP-PISA) present outputs from different cell lines and were not analyzed with the same software tools (which can also add variability –see next answer–). It allowed us to show that DIA did not reduce the sensitivity overall, but we cannot directly compare the performances of the two strategies like we do in Figure 2 and 3. Nonetheless, the DIA-PISA experiment did find around half of the staurosporine targets which were also identified in TMT-PISA experiments, even though the experiments were performed on different biological material (see venn diagram of the overlap below).

However, between the TMT-TPP and TMT-TPP-PISA experiments we can do a direct comparison. In the experiments presented in Figure 2E, all 7 Staurosporine targets identified in TMT-TPP were also identified as targets in TMT-TPP-PISA suggesting that TPP-PISA adoption of the TPP method can reproduce the same targets.

Staurosporine has been extensively used for the kinobead technology and multiple studies have developed probes based on Staurosporine for kinase panel applications. One such study determined saturating curves for over 200 kinases using Staurosporine based probe by Hirozane et al, *Bioorganic & Medicinal Chemistry Letters* 2019, (<https://www.sciencedirect.com/science/article/pii/S0960894X19305864>). In our TPP-PISA experiments (DIA and TMT combined), we found 22 of the 203 kinases that Hirozane et al. identified as Staurosporin binders (and 6 of our hits were not identified in Hirozane et al.), which could indicate on the sensitivity and false discovery proportion of the assay. Nonetheless, this is a very rough estimation, and we believe that one of the main advantages of TPP-PISA-based techniques is the ability to capture downstream effects on the proteomes, ie. indirect destabilization of non-kinases that should not be counted as false positives. Figure 5D shows that Staurosporin can have tissue-specific effects on the output of the TPP-PISA analysis, which makes it ill suited for true/false positive hit estimation in the context of this work.

If the reviewer's concern is in regards to the false discovery rate of peptide/protein identification. We used 1% FDR for the respective search software utilized (MaxQuant 1.5.7.0 for DDA and TMT analysis, and DIA-NN 1.81 for DIA). We have added this information to materials and methods for DDA and DIA respectively (see below).

“1% false discovery rate (0.01) was utilized at the peptide spectral match and protein level.”

For DIA-NN we have added the following to the materials and methods describing the search parameters.

“...and the precursor FDR (%) was set to 1.0.”

The comparison of protein ID and throughput alone (Fig. S2) is not sufficient for method benchmarking.

The reviewer is correct to note that protein identifications is not the only metric that should be presented for comparing the performance of two different search software (see previous answer). The aim of Figure S2 is to show that DIA-based analysis allows to identify a similar number of protein groups as the TMT-based strategy, which was a prerequisite to be as sensitive when evaluating the entire pipeline. That being said, comparing the number of identifications here is flawed because DIA-NN assembles and reports protein groups differently compared to MaxQuant. This is why we focused on statistically regulated staurosporine targets from similar analysis tools (with the same grouping strategies) in the main figures (Figure 1 and 2), and as other authors and researchers in the field have highlighted, including the original PISA as described in Gaetani et al (JPR, 2019). We realized that this issue of protein grouping may not be obvious to readers so we have added this cautionary note in the main text in line 161:

“From our analysis, DIA based TPP-PISA resulted in a larger number of identified proteins compared to TMT TPP and TMT TPP-PISA while utilizing only ~25% of the MS analysis time (Supplementary Figure S2A), however it must be noted that comparing protein groups across different search software can be difficult due to difference in assembly of protein groups.”

In addition, only staurosporine is used to validate the soundness of the method. Can the authors use at least two model drugs rather than just one compound to test the ability of the method to identify true drug targets?

We used Staurosporine because it is the standard compound several other researchers and authors have utilized to benchmark TPP and other chemical proteomics methods. We have added six references to other papers utilizing Staurosporine to develop chemical proteomic methods and added the following statement at line 122:

“Staurosporine is often utilized to benchmark chemical proteomic methods and a key component for the “Kinobead” technology, which is based on competitive affinity enrichment of kinases^{8,19–23} “

We agree that other compounds could similarly be utilized, however utilizing the same compound facilitates generalization across methods and comparison of the techniques across laboratories.

Although we only use Staurosporine for the direct comparisons of the sample preparation strategies, we use 22 drugs on the rat organs. Among these drugs, some have known targets that can illustrate the ability of the method to identify drug binders (among other drug-driven protein stability changes). Two compounds from our dataset SHP099 and Cobemitinib for example, have very specific known targets PTPN11 and MAP2K1/2, respectively, and these were identified as targets in our dataset.

The authors reported the DIA-TPP-PISA method using filter plates rather than centrifugal precipitation. This greatly simplifies the TPP procedure. However, it would be useful to explain why the PTFE filter with a pore size of 0.45 μm was used. Please characterise the proteins retained/eluted by the filters with different pore sizes.

Quite simply because this plate (https://www.merckmillipore.com/DK/en/product/MultiScreen-Solvinert-96-Well-Filter-Plate,MM_NF-MSRLN0410?) was the one we had utilized in the lab prior for removal of protein aggregates. It is not possible to purchase different pore sizes of this same filter plate. Generally, the 0.45 μm pore size should be sufficiently big for all soluble proteins, therefore the purpose of the plates is not necessarily to filter proteins by size, but to remove insoluble proteins. In theory the polytetrafluoroethylene chemistry of the filters should prevent non-specific interactions with proteins, and it should only filter out protein aggregates formed through heating of the soluble protein mixtures. We have added this note to the text on line 203:

“The 0.45 μm pore size is theoretically large enough for soluble proteins and protein complexes to pass through. The hydrophilic properties of the PTFE filter should prevent non-specific interactions, allowing soluble proteins to pass through while retaining large protein aggregates induced from heating of soluble protein mixtures.”

We analyzed the tissue lysates that were used as input for the TPP-PISA analysis of the rat tissues treated with staurosporine. Figure 1 below shows the distributions of the protein molecular weight of the proteins detected in the rat tissues lysates before (red) and after TPP-PISA analysis (blue). There is a slight but very limited shift towards lower molecular weight proteins after TPP-PISA, but it could be due to these proteins precipitating more due to heating.

Figure 1: Distribution of the molecular weight reported in UniprotKB for the proteins unambiguously identified in the rat tissue lysates (red) and after TPP-PISA (blue). A) Histogram of the protein groups identified when combining all the tissue samples. B) violin plot of log₁₀-transformed molecular weights for each tissue lysate with the associated median value (“+”). The protein groups containing several genes were excluded from the analysis, and we used the molecular weights reported for the reference Uniprot accessions only.

Why does the stability of staurosporine target kinases change in different tissues? Is this change related to organ-specific expression levels of the kinases or other metrics? Please analyze and discuss the reasons for this.

The reviewer poses a good question, and it is difficult to determine currently why this effect of temperature stability direction changes depending on the tissue for some kinases. It is possible that the context in which a kinase is present depends on the cellular presence/abundance of other proteins within the cell. As the expression of these proteins is different across tissues, it could be possible that under certain cellular conditions a kinase is in a different cellular context. We have added this to the discussion section with the following:

“Furthermore, it could be difficult to discern whether the proteomes of cell lines differ sufficiently enough to cover different proteome contexts. In contrast, animal models present a compelling alternative, as the inherent differences in various organs are mirrored in their proteome compositions. This was exemplified by the differential temperature stabilization direction of some kinases upon Staurosporine in different rat tissues. This could indicate more complex behavior of kinases due to organ-specific kinase abundance or that of the surrounding proteome context.”

We analyzed the rat tissue lysates used as input for the TPP-PISA analysis (also presented Figure 1 above), and focused on the kinases presented in the Figure 4D of the manuscript. The figure below shows that some kinases are not detected in all tissues, like Cdk6 that is only detected in the kidney and the spleen before (Fig. 2 here) and after TPP-PISA (Fig. 4D). Some kinases have varying intensities in the different tissues, like Aak1. This protein's stability is significantly affected in the TPP-PISA experiments of the liver, spleen, hippocampus, brain and prefrontal cortex, which are not the ones with the higher intensity in the input sample. The same observation can be done for Pak1. So although the starting amount of a protein in the input sample can affect the output of the TPP-PISA experiment, it does not seem to be directly correlated. Furthermore, these two examples show that we can catch protein destabilization with input samples where the protein is more difficult to detect.

Figure 2: Relative signal of a selected set of kinases in the rat tissue lysates before TPP analysis. Tile map of the relative log₂-transformed signal of kinases (mean of 2 injections after row-wise centering using median) in the tissue lysates before TPP-PISA analysis. The maximum log₂-transformed fold changes between tissues is presented in the bar plot on the right. "*" indicate tissues in which kinases have a q-value ≤ 0.05 (like in Figure 4D of the manuscript). Tissues and genes are ordered by hierarchical clustering.

Intriguingly, the kinase Fer is only detected in the total lysate of the hippocampus, but after TPP-PISA it is only detected (and significantly regulated) in the kidney. This indicates that the loss of complexity after TPP-PISA could increase detection of proteins that are not identified in

the data that we have at hand. Exploring fully the impact of input sample composition on TPP-PISA output would require a more in depth analysis.

Although the initial amount of protein is identical in each group, the amount of soluble protein from each sample after heating is not equivalent across temperature groups. How do the authors perform the 'cross-run normalization' correction when examining the LFQ DIA data? Does this normalisation step reduce the observed protein differences?

We agree with the reviewer that higher temperatures lead to higher amounts of insoluble protein, this is why we combine the volume of 3 temperatures (for each sample) and filter out the insoluble fraction afterwards. This way, the amount of insoluble/soluble protein across samples should be equivalent. Having said that, the reviewer is correct to note that the remaining soluble proteome complexity is reduced after heating at higher temperatures. This could be seen in the original comparison in the TMT based comparison of TPP and TPP-PISA (see figure below). Here we observed that the number of proteins identified were less in TPP-PISA experiments (orange bar) compared to the two conditions of the standard TPP (red bars). We believe this is due to the fact that since TPP also samples from lower temperatures, proteins that otherwise might be insoluble at higher temperatures are thus preserved and more likely to be detected.

Figure 3: Number of protein groups identified in the TPP and TPP-PISA experiments (related to Figure 2 of the manuscript). Bar plot showing the mean number of protein groups, with standard deviation.

Nonetheless, despite lower protein coverage, we find a higher number of Staurosporine protein targets because the method is more sensitive to temperature stabilization.

For the DIA analysis, we utilized the DIA-NN normalization option for each analysis. In DIA-NN, this was the cross-run normalization parameter which was set to “RT-dependent”. You can find below the boxplots of all the runs corresponding to the HeLa experiment with the metabolic drugs:

Figure 4: Log10-transformed DIA-NN output for the HeLa cells treated with metabolic drugs.

The DIA-NN normalization accounts for differences in the sample signal due to technical variability caused by different instances such as pipetting. As discussed above, since we compared two samples (DMSO vs drug-treated) containing a mixture of lysates heated at the same temperatures, we did not expect (or observe) global differences that would need specific normalization strategies.

Minor points.

1.The authors only included methods such as DARTS, Lip-MS and CETSA. A more comprehensive review of the current target ID literature, such as Thermal Proteome Profiling (TPP), Solvent Induced Protein Precipitation (SIP) and Target-Responsive Accessibility Profiling (TRAP), should be included.

We thank the reviewer for the suggestions and have accordingly added references to the above-mentioned methods in the introduction.

The section "Mass spectrometry analysis" lacks a description of the liquid phase gradient. Also, please add whether the LC gradient and instrumentation differ between DIA and TMT acquisitions.

We thank the reviewer for pointing out this omission, and we have added the gradient information in materials and methods under the “Mass spectrometry analysis” section. We have added the gradient information for the TMT labeled samples where the peptides were injected and separated using the EASY-nLC 1200 system:

“For TPP and TPP-PISA samples with TMT labeled peptides, a 220-minute gradient going from 5% to 25% Buffer B, this was followed by a 20-minute increase to 40% Buffer B. The column was subsequently washed by increasing Buffer B to 80% in 5 minutes where it was held

for an additional 5 minutes followed by a decrease back to 5% B in 5 minutes where it was held for an additional 5 minutes for column equilibration. For samples analyzed with a coupled EvoSep One LC system, peptides were separated on a 15cm C₁₈ analytical LC column (0.15 × 150 mm, 1.9 μm beads, EV-1106) utilizing the standardized 30 samples-per-day (SPD) method, which is a 44-minute active LC gradient with a total run time of 48 minutes. The analytical column was equilibrated at room temperature and operated at a flow rate of 500 nL/min. The gradient was obtained by flow injection of Evosep solvent A (H₂O/0.1% FA) and solvent B (acetonitrile with 0.1% FA) and peptides eluted by a linear gradient up to 35% solvent B”

This information ensures reproducibility for the LC/MS analysis..

In line 122, "10 μm staurosporine" should be corrected to "10 μM staurosporine".
Different yellow series lines in Figure 6 are difficult to discern.

We thank the reviewer for catching the error and have corrected it in the text. We have also changed the yellow series lines in Figure 6 to another color that is easier to discern.

A

B

C

Reviewer #3 (Remarks to the Author)

Batth et al. describe a Proteome Integral Stability Analysis workflow using a data-independent acquisition mass spectrometry-based proteomics protocol. By applying it to profile 22 compounds in different cell and tissue systems the authors find a previously undescribed target of the commonly used NSAID Ibuprofen. Many of the protocol adaptations that the authors describe are not new and should be referenced correctly. Moreover, while the combination of adaptations to the workflow does have advantages over existing workflows, as the authors rightfully point out, a careful comparison to existing approaches is needed to give potential future users a clear overview of what is best for their use case.

We thank the reviewer for the constructive feedback, and we have updated the text to address the reviewers points.

Major points:

- While disadvantages of TMT data-dependent acquisition such as limited multiplexing capability are discussed, advantages such as higher sensitivity are not mentioned. This point would become more evident when data independent acquisition (DIA) TPP/PISA data would be compared to a dataset derived with data-dependent acquisition (DDA) using TMT, e.g. using the Staurosporine dataset by the Savitski et al. (2014) paper which would make a nice benchmark to compare the new DIA approach with the previous DDA one. This could potentially be even extended by comparison to a newer dataset using TMT-18 by Zinn et al. (2021) Journal of Proteome Res.

We thank the reviewer for highlighting the prior datasets and need for comparisons. We agree that it is important to compare methods, which is why we performed a traditional TMT based TPP experiment using the exact same samples, as presented in Figure 2 and compared with the PISA method (on the same sample). As we note in the manuscript, our goal was to strike a balance between throughput, sensitivity, and proteome depth. However, the use of different cell lines, instruments, and methods make direct comparisons across different papers difficult. One limitation we encountered with published papers such as the original Savitsky et al (Science 2014) paper was that the authors utilized only duplicates, making the determination of statistical variance in their data difficult. In the more recent TMTpro (16-plex) paper, the authors also note that two replicates are utilized per condition with an MS analysis time of 1.1 day, resulting in the identification of 41 kinases as Staurosporine targets using their downstream bioinformatics method. This is indeed more than 23 kinase targets we identified (Figure 3D) using our optimal DIA based workflow, but it should be noted that our data was obtained in 4.5 hours of total MS run time (45 minutes per sample) and we utilized triplicates for each condition. We believe this approach allows us to determine protein targets with greater statistical confidence. It should also be noted that the Zinn et al analysis required additional steps of TMT labeling and offline fractionation, significantly increasing preparation time.

- It would be also critical to use the latest analysis methods, e.g. nonparametric analysis of response curves (Childs et al. (2019) Mol. Cell. Proteomics) to compare TPP to PISA since the melting-point comparison does not make use of the full measurements in the TPP data and would likely show comparable or higher sensitivity in identifying kinase targets from the data .

The figure below shows the output of the NPARC analysis on the TPP and/or TPP-PISA data presented in the figure 2 of the manuscript:

Figure 5: NPARC analysis of the TPP data presented Figure 2 of the manuscript. **A)** Results of the NPARC analysis performed and presented as in (Childs et al. (2019) Mol. Cell. Proteomics) (Figure 2C). The significance of the test is presented on the vertical axis ($-\log_{10}(\text{adjusted p-value})$), and the effect size is presented on the horizontal axis: difference of the sum of squared residuals for the null hypothesis and for the alternative hypothesis. The proteins highlighted in red are kinases (as reported in the Figure 2 of our manuscript). The NPARC analysis was performed on the same normalized curves as for melting point calculation with the R package NPARC v1.16.0 and default parameters, keeping only the protein groups with minimum 39

points in the treated and control condition. **B)** Same volcano plot as figure 2C of our manuscript reported here for comparison: Volcano plots showing the statistical output of the TPP-PISA experiment. $-\log_{10}(\text{p-values})$ are plotted on the vertical axis, and the \log_2 -transformed differences between the Staurosporine- and DMSO-treated conditions are presented on the horizontal axis: differences between the corrected \log_2 -transformed MS intensities of the TMT reporters. Proteins on the right side of the volcano plot were stabilized by Staurosporine while proteins on the left side were destabilized. **C)** Same plot as figure 2E with the results of the NPARC analysis: Number of kinase (red) and non-kinase (black) protein groups identified in the three experiments with $q\text{-value} < 0.05$ and $q\text{-value} < 0.01$.

Here, we can see that the TPP-PISA identifies more kinases and less non-kinases than the NPARC and the TPP experiment. Performing a fair and complete comparison of software tools for TPP data analysis would necessitate testing out different filtering strategies, statistical thresholds, use of different data sets to avoid biases, etc... This is beyond the scope of this paper. For this reason, we decided to present a straightforward comparison of the t-test strategy using either the melting points from the TPP data or the two conditions of the TPP-PISA strategy, as was presented in the original figure.

- The replacement of the centrifugation step for protein aggregate removal and also the 96 well format (at least for TPP) is not new. It has been previously applied by Selkrig et al. (2021) Mol. Syst. Biol, please rephrase and reference accordingly.

We thank the reviewer for reporting this reference. The reviewer is indeed correct that protein aggregate removal using filter plates is not new and has been used for different applications. We have altered the sentence at line 192 and added the Selkrig et al reference. It now reads:

“To address this challenge, we explored the use of filter plates as an alternative to centrifugation for removing insoluble proteins after a TPP-PISA experiment as reported earlier [**Selkrig et al reference added here**]. Filter plates have been utilized for removal of insoluble protein aggregates but their advantage over traditional centrifugation based insoluble protein removal for TPP has not been thoroughly reported.”

- Abundance vs. thermal stability in PISA: the PISA assay cannot distinguish between changes in protein abundance (also solubility which can be modulated by some compounds, especially at higher concentrations) and thermal stability. In the lysate format this is not a problem, but if potential future users would like to apply the workflow to living cells this will be an issue, please add a cautionary statement to the discussion.

We thank the reviewer for highlighting this important point. The reviewer is indeed correct to note that protein abundance may change when working with living cells in response to a drug

treatment and this can interfere with accurately discriminating between changes in stability and abundance when utilizing the PISA approach. We have added this cautionary statement to the discussion at now line 457:

“One limitation of PISA is that this method is not able to discriminate between abundance and thermal stability changes upon compound treatment when applying the workflow to living cells. This could be addressed through proteome measurement prior to the TPP temperature incubation, which could be used to normalize the abundance changes observed in the PISA measurement.”

- The authors comment on one of the described targets of Ibuprofen: “CYP2C9 was not identified as a target in our analysis” is this because the protein was not measured or because it was not found to be changed in thermal stability.

We did not detect the gene CYP2C9 in the human cell lines. We only mapped human orthologs of the proteins that were significantly regulated in minimum one rat tissue in our network analysis. Doing this, no hit was matched to the human gene CYP2C9.

To answer this comment, we looked back at the data. There is a potential rat ortholog of CYP2C9: the protein ENSRNOP00000036896 (according to our Biomart mapping, accessible in the zenodo repository). This protein is never unambiguously identified in our data where it is in the protein group “ENSRNOP00000036896.6;ENSRNOP00000067531.1” (corresponding to the rat genes "AABR07004560.1;Cyp2c13"). In the ibuprofen experiment, this protein group is exclusively quantified in the rat liver sample where it does not pass our statistical thresholds.

Overall we do not have enough evidence of the actual presence of CYP2C9 or its rat ortholog in the samples we worked with.

How are the numbers of identified protein in general comparing to DDA-based TPP datasets? Please extend discussion accordingly.

The number of protein identifications can vary depending on the cell line, but for our DIA based PISA experiments we averaged between 4600-4700 protein identifications per run (Figure S2). This was higher compared to single shot DDA-TMT TPP and PISA in our analysis, but it is less than 6894 proteins identified by Zinn et al, however this comes with the advantage of significantly decreased sample preparation and MS analysis time. We have modified the discussion and added the reference by Zinn et al. The last few sentences in the discussion section are modified to the following:

“DIA analysis resulted in a larger number of protein identifications when compared directly with DDA based TMT TPP analysis (Figure S2). However, peptide fractionation can lead to higher proteome coverage and sensitivity of TPP resulting in the identification of more drug targets [Zinn et al reference added here]. Despite these limitations, we aimed to strike a balance between throughput, sensitivity, and proteome depth with our workflow. As such, we anticipate future optimizations in thermal proteome profiling based methods that will complement other techniques in drug development and target identification.”

Minor points:

- The authors might want to acknowledge that Perrin et al. (2020) Nat. Biotechnolgy were the first to implement TPP in an in vivo and ex vivo setting of rodent organs.

We thank the reviewer for pointing out this reference and we have added this to the manuscript at line 243 in the section “EVALUATION OF RAT ORGAN EXTRACTS” with the following sentence:

“Furthermore, TPP has been implemented in vivo with mouse models and whole blood [Perrin et al reference], demonstrating the potential applicability of the method in vivo or ex vivo.”

While the code provided by the authors appears comprehensive, also providing reproduction of the manuscript figures it is difficult for potential users to navigate and try out since it mostly represents a collection of scripts, rather than a software package with easy to use functions made available for a potential user.

A GitHub page with the code and explanations of the functionality would be preferable, ideally also including unit tests of the software functions.

We completely understand this point, unfortunately developing a package is out of the scope of this work. We only made the codes available for transparency and reproducibility, this is not made for easy reuse, and it would require a lot of work to make it so (and to maintain), especially due to the variety of possible experimental set-up, data acquisition, drugs etc... that could be considered for a TPP-PISA experiment such as the one we present. Additionally, we want to point out that the analysis itself is not very complicated, and in our opinion does not require the development of a dedicated tool. We performed a classical pipeline for MS data analysis, filtering, normalization that is fully documented in the material and methods (and in the Zenodo repository), and we performed a “simple” t-test after selection of best-suited control samples.

Response to reviewers.

We thank the reviewers for their constructive feedback. Please find our point-by-point response to each of the remarks. We have highlighted our response in blue color.

Reviewer #2 (Remarks to the Author)

I feel most of the issues have been fully addressed.

We thank the reviewer for their constructive response to improve the manuscript and for taking their time to review.

Reviewer #3 (Remarks to the Author)

The authors have addressed the large majority of my comments and I recommend the manuscript in its current form for publication with a minor addition: I would like the authors to add one more half sentence clarifying that the 'standard TPP' experiment they performed (p. 6) is not done using peptide fractionation (I guess to make run time comparable to DIA), which is a crucial step in TMT-based TPP experiments to guarantee detection numbers and precision in peptide quantification and strongly influences the quality of the obtained data which in turn affects sensitivity and specificity in staurosporine target identification.

We thank the reviewer for their review and helping us improve the manuscript. We have modified the paragraph on page 6, lines 133-136.

“It should be noted that to make the analysis comparable, TPP was performed without peptide prefractionation, which is a crucial step in TMT-based TPP experiments to guarantee higher peptide detection numbers and precision in peptide quantification which can influence the sensitivity and specificity in Staurosporine target identification.”